# TMT-Opsins differentially modulate medaka brain function in a context-dependent manner

Bruno M. Fontinha[1,2☉], Theresa Zekoll[1,2☉], Mariam Al-Rawi[1,2], Miguel Gallach[3¤a‡], Florian Reithofer[1,2¤b‡], Alison J. Barker[4], Maximilian Hofbauer[1,2,5], Ruth M. Fischer[1¤c], Arndt von Haeseler[1,2,3,6], Herwig Baier[4], Kristin Tessmar-Raible[1,2,7] *

**1** Max F. Perutz Laboratories, University of Vienna, Vienna, Austria, **2** Research Platform "Rhythms of Life," University of Vienna, Vienna, Austria, **3** Center for Integrative Bioinformatics Vienna, Max F. Perutz Laboratories, University of Vienna and Medical University of Vienna, Vienna, Austria, **4** Max Planck Institute of Neurobiology, Martinsried, Germany, **5** loopbio, Vienna, Austria, **6** Bioinformatics and Computational Biology, Faculty of Computer Science, University of Vienna, Vienna, Austria, **7** FENS-Kavli Network of Excellence, Brussels, Belgium

☉ These authors contributed equally to this work.
¤a Current address: iLabSystems, Vienna, Austria
¤b Current address: IMC Fachhochschule Krems, University of Applied Sciences, Krems, Austria
¤c Current address: Boehringer Ingelheim, Vienna, Austria
‡ These authors contributed equally to this work.
* kristin.tessmar@mfpl.ac.at

**Data Availability Statement:** All relevant data are within the paper and its Supporting Information files.

## Abstract

Vertebrate behavior is strongly influenced by light. Light receptors, encoded by functional opsin proteins, are present inside the vertebrate brain and peripheral tissues. This expression feature is present from fishes to human and appears to be particularly prominent in diurnal vertebrates. Despite their conserved widespread occurrence, the nonvisual functions of opsins are still largely enigmatic. This is even more apparent when considering the high number of opsins. Teleosts possess around 40 opsin genes, present from young developmental stages to adulthood. Many of these opsins have been shown to function as light receptors. This raises the question of whether this large number might mainly reflect functional redundancy or rather maximally enables teleosts to optimally use the complex light information present under water. We focus on *tmt-opsin1b* and *tmt-opsin2*, c-opsins with ancestral-type sequence features, conserved across several vertebrate phyla, expressed with partly similar expression in non-rod, non-cone, non-retinal-ganglion-cell brain tissues and with a similar spectral sensitivity. The characterization of the single mutants revealed age- and light-dependent behavioral changes, as well as an impact on the levels of the pre-prohormone *sst1b* and the voltage-gated sodium channel subunit *scn12aa*. The amount of daytime rest is affected independently of the eyes, pineal organ, and circadian clock in *tmt-opsin1b* mutants. We further focused on daytime behavior and the molecular changes in *tmt-opsin1b/2* double mutants, and found that—despite their similar expression and spectral features—these opsins interact in part nonadditively. Specifically, double mutants complement molecular and behavioral phenotypes observed in single mutants in a partly age-dependent fashion. Our work provides a starting point to disentangle the highly complex

**Funding:** This work was supported by the research platform "Rhythms of Life" of the University of Vienna and a FWF (http://www.fwf.ac.at/) SFB grant (#SFB F78) to K.T-R and A.v.H, a FWF (http://www.fwf.ac.at/) START award (#AY0041321), research project grant (#P28970), and funding from the European Research Council under the European Community's Seventh Framework Programme (FP7/2007-2013)/ERC Grant Agreement 337011 and the Horizon 2020 Programme ERC Grant Agreement 819952 to K.T-R. A.J.B. and H.B. acknowledge funding by SFB 870 „Assembly and Function of Neural Circuits" and the Max Planck Society. R.M.F. was supported by a PhD fellowship of the Boehringer Ingelheim Foundation (https://www.bifonds.de/fellowships-grants/phd-fellowships.html) and T.Z. by an uni: docs fellowship and the doctoral school CoBeNe (https://vds-cobene.univie.ac.at/) of the University of Vienna. The funders had no role in study design, data collection and analysis, decision to publish, or preparation of the manuscript.

**Competing interests:** I have read the journal's policy and the authors of this manuscript have the following competing interests: M.H. is the CEO of loopbio GmbH, a company manufacturing and selling solutions for advanced behavioral animal tracking. All other authors declare that no competing interests exist.

**Abbreviations:** AI, avoidance index; dpf, days post-fertilization; ETO, Encephalopsin/TMT-Opsin; RGC, retinal ganglion cell; qPCR, quantitative PCR; RNAseq, RNA sequencing; RT, room temperature; TALEN, transcription activator-like effector nuclease; ZT, zeitgeber time.

interactions of vertebrate nonvisual opsins, suggesting that *tmt-opsin*-expressing cells together with other visual and nonvisual opsins provide detailed light information to the organism for behavioral fine-tuning. This work also provides a stepping stone to unravel how vertebrate species with conserved opsins, but living in different ecological niches, respond to similar light cues and how human-generated artificial light might impact on behavioral processes in natural environments.

## Introduction

Organisms are exposed to a large range of light intensities and spectral changes. While humans are well aware of the visual inputs from their environments, the range of intensity and spectral changes that animals, including humans, naturally undergo are less consciously experienced. Across the day, light intensity routinely differs by several orders of magnitude depending on the angle of the sun above the horizon and weather conditions (e.g., a $10^3$-fold difference occurs between a sunny day and the moments before a thunderstorm, which is a similar difference between a sunny day and the average office illumination) [1].

While vision evolved multiple mechanisms to compensate for these differences, allowing us to see almost equally well across about $10^6$ orders of magnitude of light intensity [1], light differences have wider effects on physiology and behavior than solely impacting vision. This has been well documented for vertebrates, including mammals [2]. Light prominently affects human (and other mammalian) physiology, mood, and cognitive function via the entrainment of the circadian clock [3,4]. In mice, it was shown that, in addition to setting the circadian clock, light information conveyed via the perihabenular nucleus also directly affects brain regions that control mood [5], while moderate exposure to UVB light promotes the biosynthesis of glutamate and results in enhanced learning and memory [6]. Furthermore, photoperiodic effects on mammalian mood, cognition, and fear behavior are well documented, which are for example connected with changes in transcript levels of *tyrosine hydroxylase* and *pre-prosomatostatin1* [7].

These far-reaching impacts of light on behavior and physiology are not just documented for mammals, but span across vertebrates. Seasonality, and in particular photoperiod, controls major hormonal changes connected to breeding, molt, and song production in birds (reviewed in [8]), while light modulates motor behavior independently of the eyes in frogs [9] and teleosts [10].

The past years have revealed that opsins, proteins mediating light sensation in vertebrates, are present in various cell types inside and outside vertebrate eyes, including specific brain neurons [11–14]. Zebrafish possess 42 *opsin* genes [15], and additional genomic and transcriptomic data suggest that *opsin* gene numbers are similarly high in other teleost species (http://www.ensembl.org).

Biochemical analyses, tissue culture assays, and electrophysiological recordings on brain tissues suggest that most, if not all, of these opsins can function as light receptors [14–18]. Light has been shown to reach deep brain regions of several mammals and birds [2,11,19]. Given the typically smaller sizes of fish brains, especially in medaka and zebrafish, it is highly conceivable that light will reach cells inside the fish brain. Based on the expression of several opsins in inter- and motor-neurons in larval as well as adult zebrafish and medaka fish brains, we suggested that these opsins could modulate information processing, depending on ambient light conditions [14].

Studies on young zebrafish larvae indeed implicate nonvisual opsins in specific light-dependent behaviors, such as the photomotor response present at 30 h post-fertilization [20], for which, however, the exact identity of the opsins involved remain unclear. The suppression of

spontaneous coiling behavior in larvae younger than 24 hours post-fertilization is mediated, for its green light component, by VAL-opsinA [21], while melanopsins have indirectly been implicated to mediate part of the dark photokinesis response (in larvae 5–7 days post-fertilization [dpf]) [13]. The latter assay was further extended to investigate search pattern strategies of 6- to 7-dpf larvae under extended periods of sudden darkness. In this assay Cas9/Crispr-engineered *opn4a* mutants showed altered local search behavior [22]. Combinational analyses of enucleated wild-type fish, *otpa* mutants, and enucleated *otpa* mutants indicated a contribution of brain-expressed *opn4a* [22]. However, as there was no comparison between enucleated *opn4a* mutant and wild-type fish, the contribution of *opn4a* expressed in the eye versus the brain [23] remained unresolved.

While these studies have started to provide insights into the understanding of nonvisual/ deep brain opsin functions, several important biological questions have remained entirely unaddressed.

First, nonvisual opsins are not just present during young larval stages, and their later functions remain unclear. How do the early functions compare to functions during later juvenile stages, when the nervous system has further grown and differentiated?

Second, another line of evidence suggests that nonvisual opsins are involved in conveying light information for chronobiological functions, such as the circadian clocks autonomously present in different fish tissues [24,25]. Outside the tropical/sub-tropical zebrafish, nonvisual opsins have also been suggested to convey photoperiod information, particularly to the reproductive system [26]. Is there evidence for such a function in a species for which seasonal cues are naturally relevant?

Third, one of the maybe most puzzling questions concerns the number of opsins in teleost fish. Why are there so many? Is it simply redundancy of the system, or do these opsins control physiology and/or behavior in a more complex, possibly synergistic manner?

Here we focus on the functional characterization of 2 members of the Encephalopsin/ TMT-Opsin (ETO) family, which is characterized by a particularly low rate of sequence changes over time and represents ancestral c-opsins [14,27]. One of its subfamilies (Encephalopsin) is conserved up to placental mammals [14,28,29]. We focused our analyses on medaka fish (*Oryzias latipes*), since these fish show a light-dependent seasonal breeding response as an adaptation to the photoperiod changes naturally occurring in their habitat [30,31]. We decided to specifically focus on TMT-Opsin1b and TMT-Opsin2. Spectral analyses in cells and on purified proteins suggest that these opsins have a similar wavelength preference, while also being expressed in close vicinity to each other or even partly overlappingly ([14,32] and this study), thus providing an interesting entry point for testing complementing versus non-complementing functions. We generated medaka *tmt-opsin1b* and *tmt-opsin2* mutants, and analyzed mutant and sibling wild-type fish in multiple behavioral and molecular assays. Our functional work on even just 2 nonvisual opsins already indicates a complex, (at least in part) non-redundant light information processing system that modulates a neurohormone and a voltage-gated sodium channel subunit, as well as behavior. In addition, our work provides several examples of how relatively small differences (in age and light intensity) manifest themselves in significant behavioral changes.

## Results

### *Ola-tmt-opsin1b* mutants exhibit light-dependent altered avoidance responses

We started our investigation with medaka *tmt-opsin1b*, by generating several independent mutant alleles using transcription activator-like effector nuclease (TALEN) technology

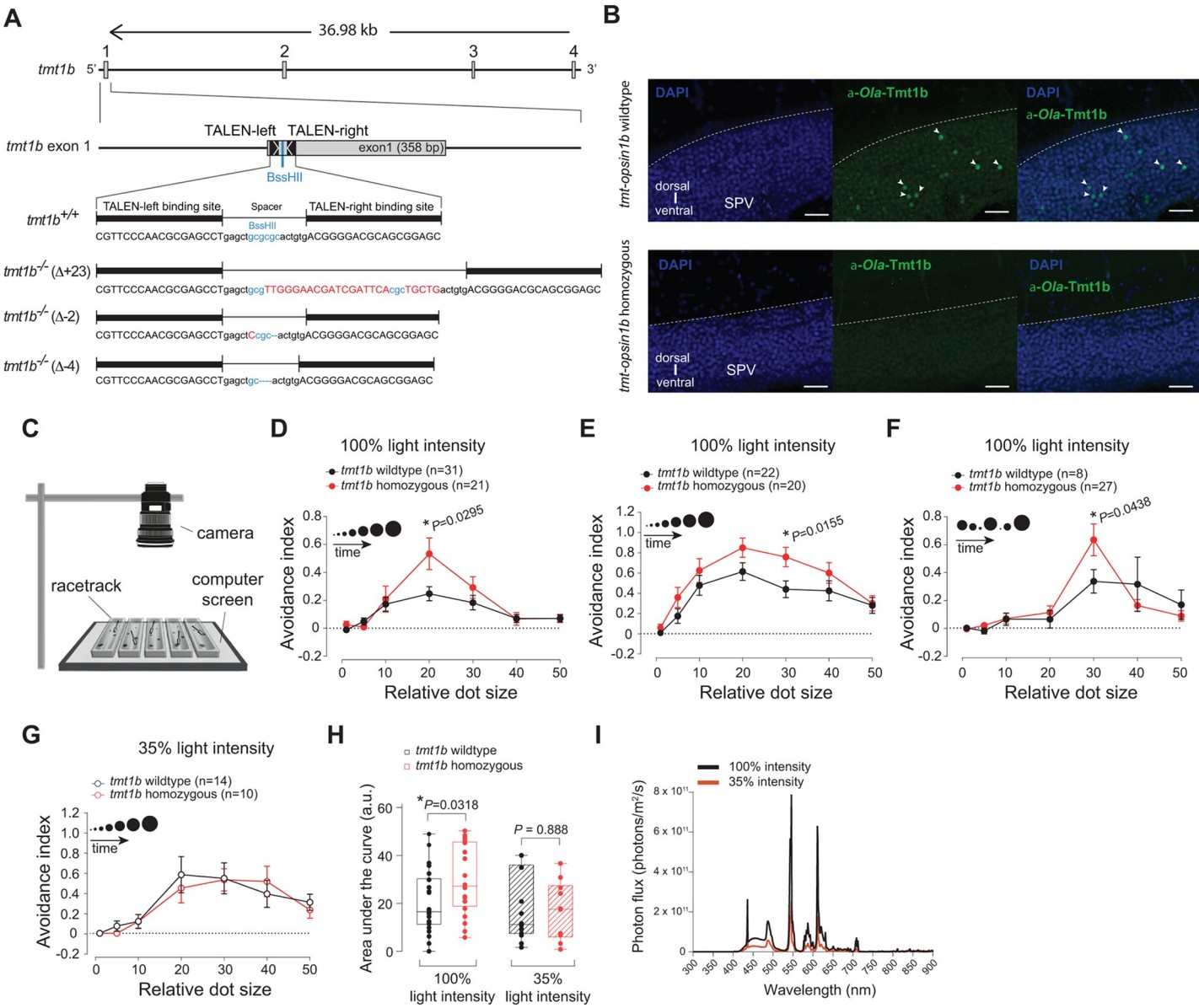

**Fig 1. Mutations in *Ola-tmt-opsin1b* cause light-dependent differences in an avoidance response test.** (A) Genomic locus of the *Ola-tmt-opsin1b* (abbreviated in figure as *tmt1b*) gene and corresponding mutant alterations. Transcription activator-like effector nuclease (TALEN) binding sites: black boxes; exons: gray boxes; inserted/substituted nucleotides: red; deleted nucleotides: dashes. (B) Confocal images of anti-*Ola*-TMT-Opsin1b (a-*Ola*-Tmt1b; green) and nuclear DAPI (blue) staining in wild-type and *tmt-opsin1b* (Δ-2/Δ-2) mutant coronal tectal slices. SPV: stratum periventriculare. Scale bars: 20 μm. (C) Schematic of the behavioral setup. (D) Size discrimination tuning curve of larvae reacting to moving dots presented in ascending size order. (E) Behavioral analysis as in (D), performed in a different laboratory with a different computer screen and surrounding (light intensity = 2.2 × 1,013 photons/m$^2$/s). (F) Avoidance responses to the dots displayed in a pseudo-random order. (G) Avoidance responses in the ascending dot size paradigm with background light reduced to 35% of its initial intensity (35% light intensity = 0.77 × 1,013 photons/m$^2$/s). (H) Total avoidance (area under the curve) under different light intensities. (I) Spectra of the emitted computer screen light used for the 100% (black) and the 35% (orange) light intensity. Data presentation: (D–G) mean (± SEM); (H) single bar: median; box: first and third quartile; whiskers: minimum and maximum; all assays done with larvae 9–12 days post-fertilization. a.u.: arbitrary units. Metadata: S1 Data. For further details on mutants, responses to contrast changes, spectra, and mutant versus wild-type development: S1–S3 Figs.

(Fig 1A). All 3 mutant alleles are predicted to result in an N-terminally truncated protein, prior to the first transmembrane domain (S1A–S1D and S1G Fig) and are thus likely functionally null mutants. We confirmed the absence of functional protein with a previously generated antibody against *Ola*-TMT-Opsin1b [14] (Fig 1B). We next tested for noticeable

developmental alterations by detailed staging of the developing larvae according to morpho-
logical criteria [33]. We did not observe any difference between wild-type and homozygous
mutant larvae (S2A–S2C Fig). We also crossed a strain expressing GFP under the control of an
*ath5* enhancer fragment in the developing retinal ganglion cells (RGCs) [34,35] with the *tmt-
opsin1b*$^{-/-}$ fish. The resulting heterozygous GFP$^+$ carriers were incrossed and their larval off-
spring analyzed for any deviations in the onset of GPF expression (indicative of the timing of
RGC differentiation), axonal outgrowth, and connectivity to the tectum (indicative of RGC
and tectal development). After hatching (7–9 dpf), the fish were genotyped. As we again did
not observe any differences between mutant and wild-type fish (*n*: wild-type = 19, heterozy-
gous mutant = 27, homozygous mutant = 18; S2D–S2G Fig), we concluded that eye and tectal
development is unaffected by the *tmt-opsin1b* mutation.

Given that *Ola*-TMT-Opsin1b is expressed in the medaka optic tectum and reticular forma-
tion [14] (Fig 1B and see below), we started our functional analysis with an assay that requires
these structures. It has been shown that the responses of frogs and fish to approaching stimuli
of different sizes specifically require the tectum and reticular formation [36–40]. We adapted
an assay in which the response of individual animals to displayed moving dots of different
sizes (mimicking a range of potential prey and predator stimuli) is recorded and analyzed
(here referred to as "avoidance assay"; Fig 1C; S1 and S2 Videos). Such avoidance assays have
been used for toads [36], larval *Xenopus* [37], mice [41], goldfish [42], and larval zebrafish [40].
Medaka larvae of different stages were collected in the morning (zeitgeber time [ZT] 2–5) and
subjected to the avoidance assay. Newly hatched medaka larvae are immediately free-swim-
ming and might best be compared to zebrafish larvae at about 7–8 dpf (constantly raised at
28˚C), while later obvious developmental steps proceed similarly to zebrafish. Based on the
responses of the larvae to the moving dots, we calculated an avoidance index (AI) for each dot
size: AI = the total number of avoidances minus the total number of approaches, divided by
the total number of dots presented. Each dot size was presented 6 times to each fish, and care
was taken to use different mutant alleles in the same trial (S1H Fig).

After blind response scoring and subsequent genotyping, we observed that *tmt-opsin1b*
mutant fish exhibited a significantly elevated avoidance response compared to their wild-type
siblings, particularly to dot sizes 20 degrees ($P = 0.0295$, unpaired *t* test with Welch's correc-
tion; $t = 2.3$; df = 26.4) and 30 degrees ($P = 0.0155$, unpaired *t* test; $t = 2.5$; df = 40), decreasing
gradually towards bigger dots (Fig 1D and 1E; S1 Data). Shuffling dot size order resulted in an
overall shift of the response curve, presumably due to habituation of the fish to the larger dots
that appeared earlier (compare Fig 1D and 1F; S1 Data). Importantly, the elevated response of
*tmt-opsin1b* mutants versus wild-type siblings was maintained.

Given that different lab environments resulted in slightly different response curves of the
mutant fish (Fig 1D and 1E; S1 Data) and that TMT-Opsin1b should function as a light recep-
tor in wild-type fish, we next tested if the observed elevated response in the avoidance assay
would be altered by changes of light conditions. The difference in avoidance response between
wild-type fish and *tmt-opsin1b* mutants at 100% light intensity vanishes when the light inten-
sity is reduced (Fig 1G and 1H; $P = 0.0318$, unpaired *t* test; $t = 2.2$; df = 40, without changing
the spectrum; Fig 1I; S1 Data). Specifically, when we lowered the light intensity of the white
background projected by the computer screen to 35% of the original light intensity, the mutant
AI curve and the wild-type AI curve (compare Fig 1E, 1G and 1H; S1 Data) became statistically
indistinguishable ($P = 0.888$, unpaired *t* test; $t = 0.14$; df = 22). We also tested whether the
observed changes in the behavioral responses observed between the different light intensity
conditions might be due to contrast differences. For this, we analyzed the AI of Cab wild-type
fish under different contrast conditions (S1I Fig; S7 Data) by testing the fish for their responses
to dots of increasing luminance (keeping dot size constant at 20 degrees). Thirty-five percent

lower light levels at the computer screen correspond to a Michelson contrast of 0.93, a contrast that does not cause any observable behavioral differences compared to full light levels (S1I Fig; S7 Data), making a change of contrast an unlikely explanation for the observed drop in the AI of *tmt-opsin1b* mutant fish at 35% ambient light intensity. These results suggest that *tmt-opsin1b* normally mediates constant responses, even under changing light conditions (for comparisons of the light spectra to natural light levels see S3 Fig).

## *Ola-tmt-opsin1b* mutants exhibit strong age-dependent behavioral differences to wild-type in response to sudden light/dark changes

The avoidance assay requires the tectum for proper responses, but it also requires normally functioning eyes. Like probably all opsins, *tmt-opsin1b* is expressed in the eye, specifically in the amacrine layer [14]. We therefore next decided to use assays that do not necessarily require eyes. Animals typically respond to changes in ambient illumination with rapid changes in movement [43]. Changes in motor behavior in response to light in blinded and pinealecto-mized minnows [10], eels [44], and lamprey tails [45] provided the first evidence for extraret-inal, extrapineal "photomotor" behavior in vertebrates. Furthermore, in zebrafish larvae, short periods of sudden darkness result in an increased overall activity, which has been interpreted as "light-seeking behavior" and termed "dark photokinesis" [13,46]. Zebrafish larvae lacking the eyes and pineal organ still react to a sudden loss of illumination with an elevated locomotor activity and an undirected light-seeking behavior [13]. After a few minutes of continued dark-ness, zebrafish will subsequently decrease their amount of swimming, resulting in less distance moved during the remaining darkness time [13,47].

We first evaluated the responses of free-swimming *tmt-opsin1b* mutant juveniles (20–22 dpf) to sudden light changes. Mutant versus wild-type medaka were subjected to 30 min each of white light and dark intervals (repeated 3 times; spectra: S3E and S4A Figs), while the move-ment of the larvae was tracked automatically and evaluated using Noldus EthoVision XT soft-ware. Both *tmt-opsin1b* mutant and wild-type fish changed their swimming distance depending on the light/dark condition (Fig 2A and 2B; S2 Data), and there was a significant difference between both during the dark phases (Fig 2A and 2B; S2 Data). As there are more than 40 opsins present in teleosts, we next wondered if this phenotype became stronger under light conditions that are more restricted to the maximal sensitivity of TMT-Opsins. TMT-Op-sins are functional photoreceptors with a maximal sensitivity in the blue light range (about 460 nm [14,32,48,49]). Thus, we next modified the assay by using monochromatic blue light (for spectra see S3F and S4A Figs). Under these light conditions *tmt-opsin1b* mutant fish exhibited a clear increase of activity during the light and dark phases of the assay compared to wild-type (Fig 2C and 2D; S2 Data). Given this clear difference between white and blue light conditions, we next wondered if this response is specific to blue light. We assayed juvenile fish under lower intensity monochromatic blue, green, and red light at as similar photon numbers as pos-sible. Under these conditions we find that juvenile wild-type and *tmt-opsin1b*$^{-/-}$ mutant fish assayed under blue light, but not under green or red light, exhibited statistically significant dif-ferences (Fig 2E–2G, 2H, 2J and 2L; S2 Data; for spectra see Figs 2I, 2K, 2M and S3G–S3I). These results show that the response is specific and provides additional evidence for the blue light sensitivity of TMT-Opsin1b.

Many behavioral and brain imaging experiments with genetically altered and/or transgenic zebrafish (as a major functional vertebrate model system in neuroscience) are typically per-formed at early developmental stages when the fish have just started to swim freely (7 dpf) or even earlier (e.g., [50,51]). As detailed in the Introduction, also all analyses of nonvisual photo-receptor responses in teleosts have so far only been performed at stages not older than 7 dpf,

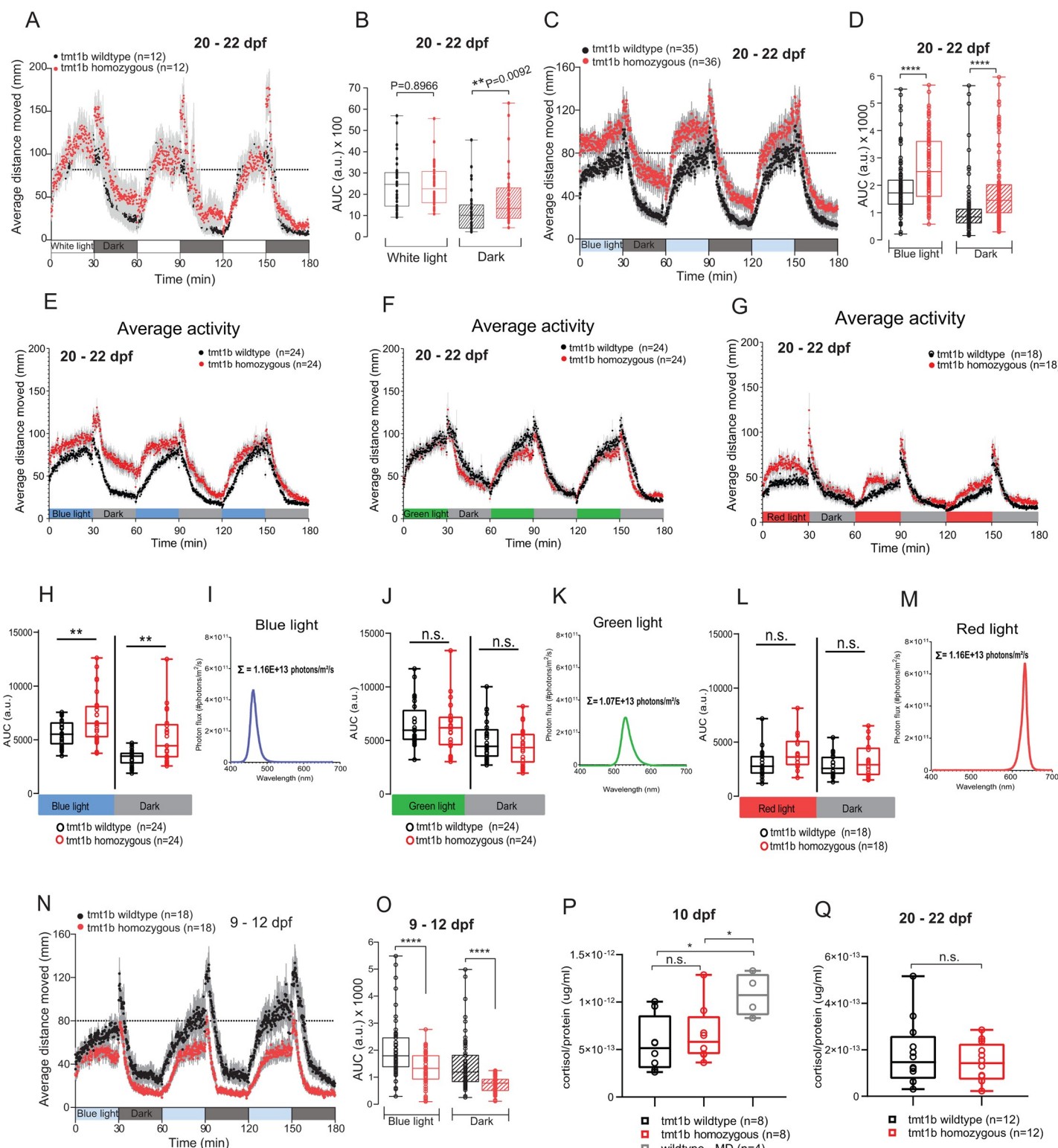

**Fig 2. Mutations in *Ola-tmt-opsin1b* impact on activity levels upon light changes in a partly age-dependent manner.** (A–G, H, J, L, N, O) Average distance moved and corresponding statistical analysis during alternating 30 min of white light or monochromatic light/dark intervals from *tmt-opsin1b* (abbreviation in figure: tmt1b) homozygous mutant (red) and wild-type (black) juvenile fish (A–G, H, J, L) and larvae (N and O). Each data point (A, C, E–G, N) represents the mean (± SEM) distance moved for the preceding 10 s. Colored boxes on *x*-axis indicate light condition. (B, D, H, J, L, O) Locomotor activity measured as area under the curve (AUC) during the entire light versus dark intervals of the trial. ****$P \leq 0.0001$; **$P \leq 0.01$. (I, K, M) Plotted spectra of the different light conditions used in (E–G, H, J, L). The sum of

photons is shown next to the respective spectrum. (P, Q) Cortisol levels measured as cortisol (μg/ml) over total protein levels (μg/ml) under baseline and mechanically disturbed (MD) conditions from larvae (P) and juvenile fish (Q). *$P \leq 0.05$. Box plots: single bar: median; box: first and third quartile; whiskers: minimum and maximum; n.s.: no significance; a.u.: arbitrary units; dpf: days post-fertilization. Metadata: S2 Data. For further details on spectra and age-dependent differences: S3 and S4 Figs.

and in most cases significantly younger [13,20,21,22]. However, it is also obvious from multiple studies that significant improvements of sensory systems and behavior occur during subsequent development (e.g., [52,53]).

We thus next tested how our results with juvenile fish compare to those at stages just after hatching. The result was strikingly different. While there was also clearly a difference compared to wild-type, instead of swimming more, mutant young larvae (9–12 dpf) swam less than their wild-type siblings (Fig 2N and 2O; S2 Data). This behavioral difference changed gradually during the days following hatching (S4B–S4I Fig; S8 Data), emphasizing that any molecular and cellular mechanism identified to control behavior has to be seen in connection with the developmental age of the tested organism.

Analyses of the "fast dark photokinesis" response rate (first 2 min after light cessation) revealed no difference in the amount of the response between *tmt-opsin1b* mutant and wild-type fish of either age, when the overall lowered or heightened mutant baseline level is accounted for (S4J and S4K Fig; S8 Data).

Finally, in order to test if the observed elevated locomotion levels are caused by generally altered stress or anxiety levels, we measured total cortisol levels. We observed no difference between mutant and wild-type siblings at young larval or juvenile stages (Fig 2P and 2Q; S2 Data), while our positive control, larvae exposed to mechanical disturbances, showed a significant increase of cortisol levels (Fig 2P; S2 Data). This strongly suggests that the difference in locomotion apparent between mutants and wild-type is indeed mediated by the acute light changes involving *tmt-opsin1b*.

### *Ola-tmt-opsin1b* mutants exhibit altered, partly eye-independent, daytime activity levels

Given the relatively consistent changes over the entire period of the experiment, we next tested the free-swimming of *tmt-opsin1b* mutants across 2 consecutive days (blue light/dark, 16 h/8 h). *tmt-opsin1b* homozygous mutant larvae and juvenile fish swam significantly more during the 1.5-h period immediately after the lights went on (Fig 3A–3D; S3 Data). This difference was reduced or disappeared when the light conditions remained stable over the rest of the day (Fig 3A and 3C; S3 Data).

Light is an important entrainment cue for the circadian oscillator. We thus next analyzed whether the mutation in *tmt-opsin1b* impacts on the phase or period length of the medaka circadian core clock. We selected 2 representative core circadian clock genes, *per1b* and *reverb*β [54], and tested their transcript oscillations in *tmt-opsin1b* mutant and wild-type larvae under light/dark and constant dark conditions. The timing of daily minima and maxima was indifferent between mutant and wild-type larvae, as was the overall cycling, clearly indicating that the *tmt-opsin1b* mutation has no impact on circadian phase or period length (Fig 3E and 3F; S3 Data).

Finally, we tested whether the increase in swimming upon sudden illumination would also occur in fish without eyes, thereby testing for a possible functional contribution of *tmt-opsin1b* outside the eye (and pineal organ, as *tmt-opsin1b* is not expressed in the pineal organ [14]). Fish were enucleated and left to recover for 1 week before the trial onset. Importantly, the differences between *tmt-opsin1b* fish and their wild-type counterparts were still observable:

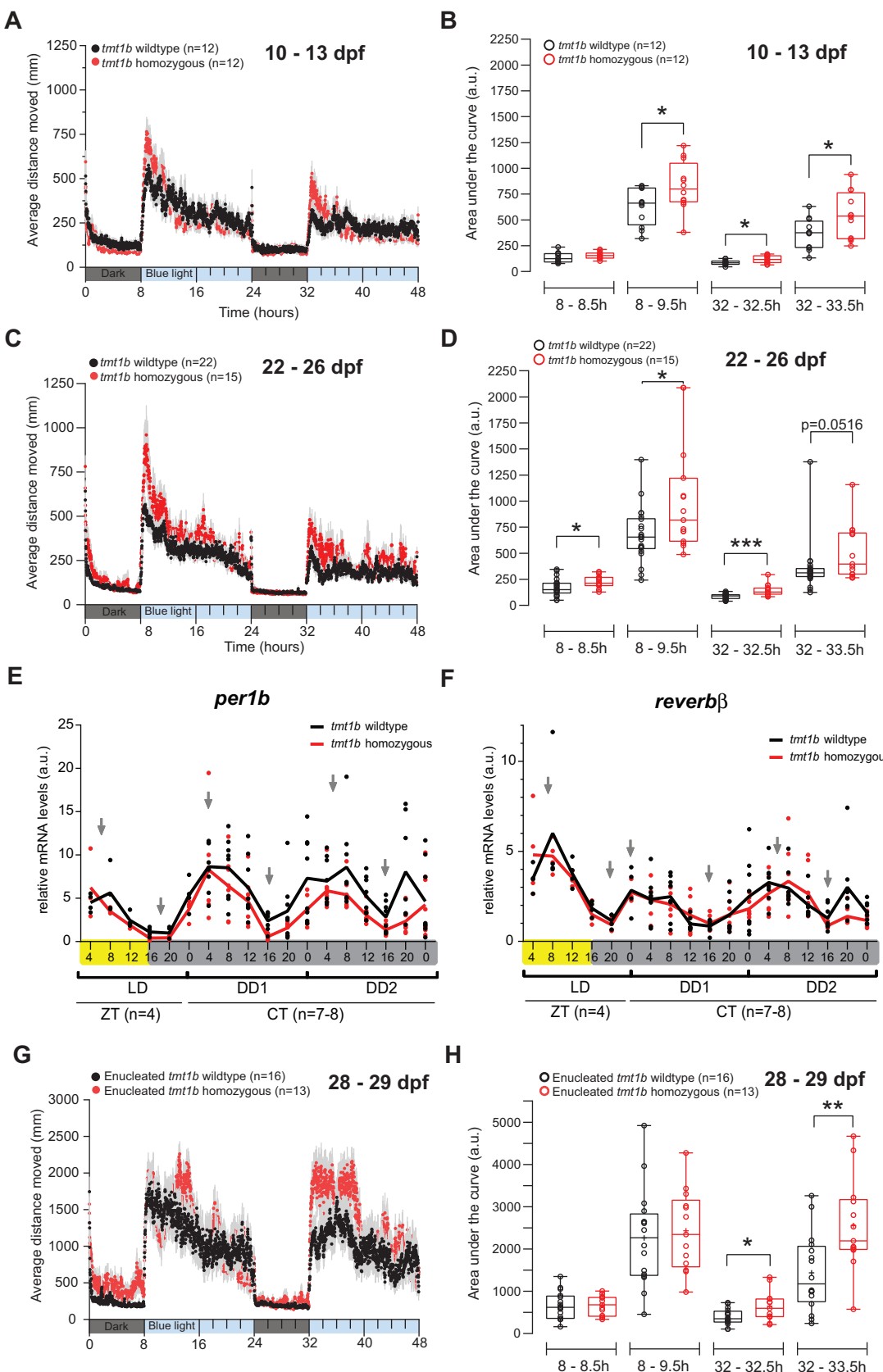

**Fig 3. *tmt-opsin1b* affects the amount of daytime rest, independently of eyes and the circadian core clock.** (A, C, G) Average distance moved during 8 h dark/16 h blue light periods for 2 consecutive days in *tmt-opsin1b* (tmt1b) mutant (red) and wild-type (black) larvae (A), juvenile fish (C), and enucleated juvenile fish (G). Each data point represents the mean (± SEM) distance moved for the preceding 1 min. Colored boxes along the *x*-axis represent light condition. (B, D, H) Locomotor activity (measured as area under the curve) during the first 30 min and 90 min after light is turned on during the 2-d trial. (B) $^{*}P_{(8-9.5\ h)}$ = 0.0331; $^{*}P_{(32-32.5\ h)}$ = 0.0247; $^{*}P_{(32-33.5\ h)}$ = 0.0323. (D) $^{*}P_{(8-8.5\ h)}$ = 0.0330; $^{*}P_{(8-9.5\ h)}$ = 0.0299; $^{***}P_{(32-32.5\ h)}$ = 0.0002. (F) $^{*}P_{(32-32.5\ h)}$ = 0.0192; $^{**}P_{(32-33.5\ h)}$ = 0.0056. (E, F) Relative mRNA levels of *per1b* (E) and *reverb*β (F) measured from whole larvae entrained to 16 h/8 h white light/dark cycles. Yellow and gray bars: light versus dark conditions during respective sampling. Gray arrows indicate the peaks and troughs of the curve. Box plots: single bar: median; box: first and third quartile; whiskers: minimum and maximum; n.s.: no significance; a.u.: arbitrary units; dpf: days post-fertilization; LD: light/dark; DD: constant dark; ZT: zeitgeber time; CT: circadian time. Metadata: S3 Data. For further details on spectra and individual, more time-zoomed movement plots: S3 and S5 Figs.

While the trend was the same during both days, it reached statistical significance only during the second day (Fig 3G and 3H; S3 Data). We thus conclude that *tmt-opsin1b*, at least in part, functions outside the eyes and modulates medaka behavior in response to different environmental light changes independently of the circadian clock.

## *Ola-tmt-opsin1b* and *Ola-tmt-opsin2* mutants exhibit additive and nonadditive responses to changes in environmental light

Teleosts such as zebrafish and medaka possess more than 40 opsins. Systematic analyses in zebrafish showed that most exhibit expression in the brain [15]. This raises the question if these opsins might just function redundantly or rather in a complex, non-redundant manner.

We purposely started to approach this question by investigating an additional opsin with similar characteristics as TMT-Opsin1b. Specifically, TMT-Opsin2 is an evolutionarily conserved ETO relative of TMT-Opsin1b, with highly similar spectral sensitivity and absorbance characteristics [14,32] and expression in adjacent, possibly partly overlapping, domains in the mid- and hindbrain [14] (Fig 4A–4E).

We thus wondered, if mutating *tmt-opsin2* would reveal redundant or synergistic functions with *tmt-opsin1b*. Using TALENs, we generated a large deletion in the *tmt-opsin2* gene (Fig 4F), removing the first 2 transmembrane helices and resulting in a dysfunctional protein (S1E–S1G Fig). Larval and juvenile *tmt-opsin2* mutant and wild-type sibling fish were assessed for their responses during the light/dark phases across 2 d, as described above.

Under these conditions, *tmt-opsin2$^{-/-}$* mutants did not display significant phenotypes, but, unexpectedly, *tmt-opsin1b/tmt-opsin2* double homozygous mutants displayed phenotypes different from the *tmt-opsin1b$^{-/-}$* single mutants. In larval stages, adding the *tmt-opsin2* mutant to the *tmt-opsin1b* mutant resulted in a complementation of the *tmt-opsin1b* phenotype during the beginning of the light phases (Figs 3A, 3B, 4G and 4H; S3 and S4 Data; for individual, high-temporal-resolution graphs see S5A–S5J Fig), but the combination of these mutations led to additive phenotypes in the same experimental test during juvenile stages (Figs 3C, 3D, 4I, and 4J; S3 and S4 Data; for individual, high-temporal-resolution graphs see S5A–S5J Fig; compare position of purple bar graphs—also indicated by arrows—to the corresponding wild-type and single mutant graphs in Fig 4H and 4J; S4 Data).

These data suggest 2 main conclusions. First, the fact that the *tmt-opsin2* mutation can compensate for *tmt-opsin1b* loss shows that these photoreceptors do not function "just" redundantly. It suggests that light information received by the fish is fed into a complex light processing system, and the output for behavior is not simply the sum of the input of all possible light receptors. Second, the functional interaction of the different opsin-based photoreceptors has a clear age-dependent component.

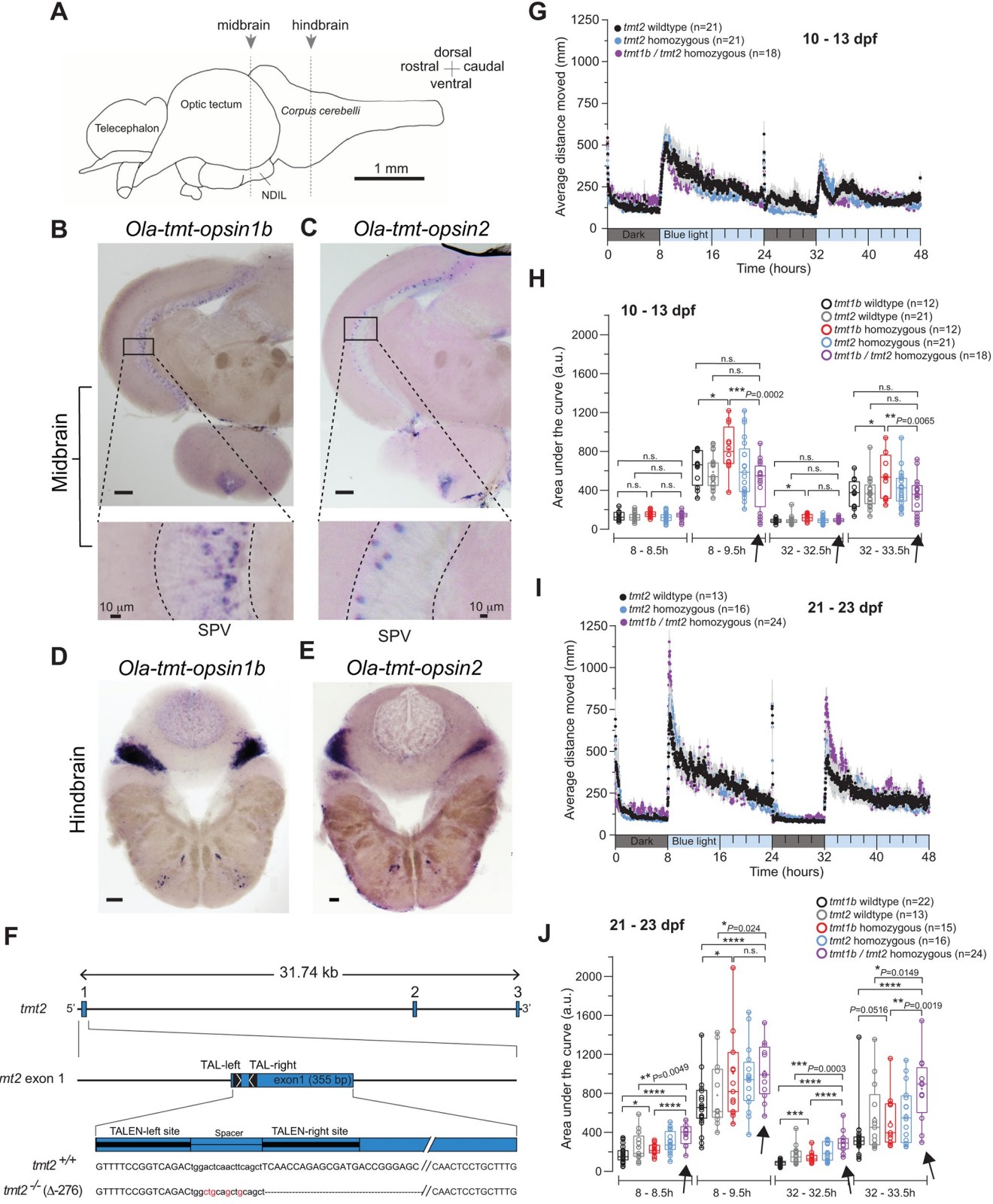

**Fig 4. Individual and combined differential impact of *Ola-tmt-opsin1b* and *Ola-tmt-opsin2* on daytime rest levels.** (A) Schematized lateral view of an adult medaka brain. Dashed lines: position of sections in (B, C) and (D, E). NDIL: nucleus diffuses of lobus inferioris (of hypothalamus). (B, C, D, E) In situ hybridization (ISH) of *tmt-opsin1b* (B, D) and *tmt-opsin2* (C, E) on coronal sections from Cab wild-type fish. SPV: stratum periventriculare. Unlabeled scale

bars: 100 μm. For detailed anatomical annotations see [14]. (F) Genomic locus of the *Ola-tmt-opsin2* (abbreviation in figure: *tmt2*) gene and corresponding mutant. Transcription activator-like effector nuclease (TALEN) binding sites: black boxes; exons: blue boxes; inserted/substituted nucleotides: red; deleted nucleotides: dashes. (G, I) Profile of average distance moved during a 2-consecutive-day trial on a 16 h/8 h blue light/dark photoperiod of (G) 10–13 d post-fertilization (dpf) larvae and (I) 21–23 dpf juvenile *tmt-opsin1b (tmt1b)* and *tmt-opsin2 (tmt2)* mutant and wild-type fish. Each data point represents the mean (± SEM) distance moved for the preceding 1 min. Colored boxes along the *x*-axis represent light condition. (H, J) Locomotor activity (assessed by area under the curve) during the first 30 min or 90 min after light is turned on in the 2-d trial of (H) larvae and (J) juvenile fish. Arrows highlight the data from the double mutants, in comparison to the respective single mutants and wild-type fish. For all panels: $^*P \leq 0.05$; $^{**}P \leq 0.01$; $^{***}P \leq 0.001$; $^{****}P < 0.0001$. Box plots: single bar: median; box: first and third quartile; whiskers: minimum and maximum; n.s.: no significance; a.u.: arbitrary units. Metadata: S4 Data. For further details on mutants and spectra, and individual, more time-zoomed movement plots: S1, S3, and S5 Figs.

## Mutations in *tmt-opsin1b* and *tmt-opsin2* induce transcriptional changes that impact on neuronal information transmission

We next wondered which molecular changes can be detected in the brain of the fish missing specific photoreceptors. It had previously been shown that differences in photoperiod cause changes on the transcript level in the rat brain, resulting in differences of neurotransmitter abundance [7]. We thus reasoned that quantitative RNA sequencing (RNAseq) might be a possible strategy to obtain unbiased insight into the changes that occur due to lack of *tmt-opsin1b* or *tmt-opsin2* function.

We sampled 3 separate brain regions (eyes, forebrain, and mid-/hindbrain; Fig 5A) at ZT 2–3 under a 16 h/8 h white light/dark regime and sequenced stranded cDNA. It should be noted that for technical reasons part of the mid-/hindbrain sample includes a portion of the posterior hypothalamus, i.e., forebrain. The resulting sequences were mapped to the medaka genome (Ensembl version 96); reads mapping to annotated exons were quantified using edgeR (Bioconductor version 3.9) [55]. The resulting tables are included as S10 and S11 Data. When comparing *tmt-opsin1b* mutants and their wild-type siblings, one differentially regulated transcript caught our particular attention, the preprohormone *sst1b* (ENSORLG00000027736: also named *sst3* or *cort*) in the mid-/hindbrain (Fig 5B; S5 Data), a member of the somatostatin/corticostatin family. We next independently confirmed the quantitative RNAseq results using quantitative PCR (qPCR) (Fig 5C, black versus red boxes for mid- and hindbrain; S5 Data). In this set of experiments, we separated the midbrain from the hindbrain tissue in order to obtain a more differentiated picture of the regulation. Given the differential effects of *tmt-opsin1b/tmt-opsin2* double mutants on behavior, we next wondered how the presence of both mutations would affect *sst1b* regulation. Adding the *tmt-opsin2* mutation compensated the downregulation of *sst1b* levels present in the *tmt-opsin1b* single mutant (Fig 5C, purple boxes compared with others for mid- and hindbrain; S5 Data). In an analogous approach, we identified the voltage-gated sodium channel subunit *scn12aa* as significantly regulated by the *tmt-opsin2* mutation (Fig 5D; S5 Data; phylogenetic tree: S8 Fig). Again, while clearly visible in the single mutant (Fig 5E, gray versus blue boxes for mid- and hindbrain; S5 Data), the effect was compensated for in the *tmt-opsin1b/tmt-opsin2* double mutants (Fig 5E, purple versus blue boxes for mid- and hindbrain; S5 Data).

These molecular data further support the notion that the loss of function of nonvisual opsins does not necessarily lead to a simple summation effect originating from the single mutants. Again, this strongly corroborates the notion of non-redundant, complex light information processing, which can modulate the neuronal function on the level of neuropeptides (*tmt-opsin1b* regulating *sst1b*) and voltage-gated channels (*tmt-opsin2* regulating *scn12aa*).

## The regulation of *sst1b* via *tmt-opsin1b* is non-cell-autonomous

In order to gain deeper insight into the connection between *sst1b* expression changes in the mid-/hindbrain, *tmt-opsin1b*, and responses to light, we next analyzed the spatial expression of

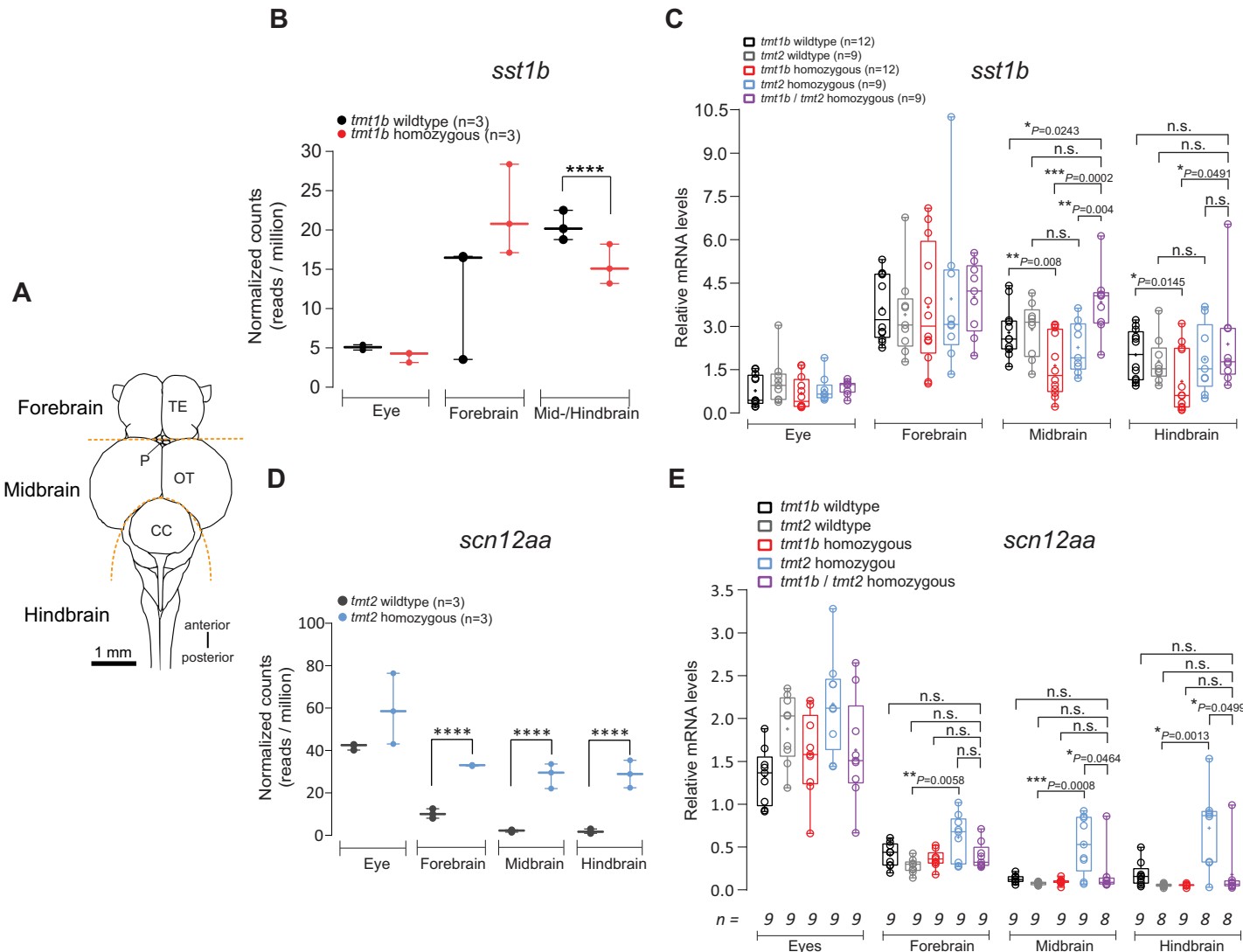

**Fig 5. *Ola-tmt-opsin1b/2* double mutants are compensated for differential transcript changes present in *Ola-tmt-opsin1b* and *Ola-tmt-opsin2* single mutants.** (A) Schematic drawing of the adult medaka brain in dorsal view. Dotted lines indicate the dissection boundaries (TE: telencephalon; P: pineal organ; OT: optic tectum; CC: corpus cerebelli). Note: part of the posterior hypothalamus is dissected with the midbrain, as it is attached to its ventral part and technically unfeasible to reliably separate. (B) Normalized transcript read counts (****adjusted $P$ value < 0.0001) for *sst1b*. (C) mRNA transcript levels for *sst1b* determined by quantitative PCR in brain areas and eyes of *tmt-opsin1b (tmt1b)* mutant, *tmt-opsin2 (tmt2)* mutant, and corresponding wild-type and double-mutant fish. Decreased *sst1b* levels are unique to *tmt-opsin1b$^{-/-}$* mutants and are compensated by the *tmt-opsin2$^{-/-}$* mutant. *$P \leq 0.05$; **$P \leq 0.01$; ***$P \leq 0.001$; n.s.: no significance. (D) Normalized transcript read counts (****adjusted $P$ value < 0.001) for *scn12aa*. (E) mRNA transcript levels for *scn12aa* in brain areas and eyes. Increased levels of *scn12aa* are unique to *tmt-opsin2$^{-/-}$* mutants, and compensated for when both *tmt-opsin1b* and *tmt-opsin2* are jointly mutated. *$P \leq 0.05$; **$P \leq 0.01$; ***$P \leq 0.001$; n.s.: no significance. Box plots: single bar: median; box: first and third quartile; whiskers: minimum and maximum; each displayed data point corresponds to 1 individual biological replicate. Metadata: S5 Data, S10 Data, S11 Data.

*sst1b*. In the midbrain, *sst1b* is expressed in several highly specific clusters of cells, none of which overlap with *tmt-opsin1b* expression (compare Fig 6A and Fig 4B; for detailed anatomical annotation of *sst1b* expression see S6 Fig; for *tmt-opsin1b* expression see [14]). This is particularly obvious for the tectum, in which *sst1b*$^+$ cells are consistently located more dorsally within the stratum periventriculare (SPV) than *tmt-opsin1b*$^+$ cells, and close to or possibly overlapping with *tmt-opsin2*$^+$ cells (compare Fig 6A with Fig 4B; anatomical annotation: [14], S6 Fig). Similar in the reticular formation of the hindbrain, likely all of the *sst1b*$^+$ cells are

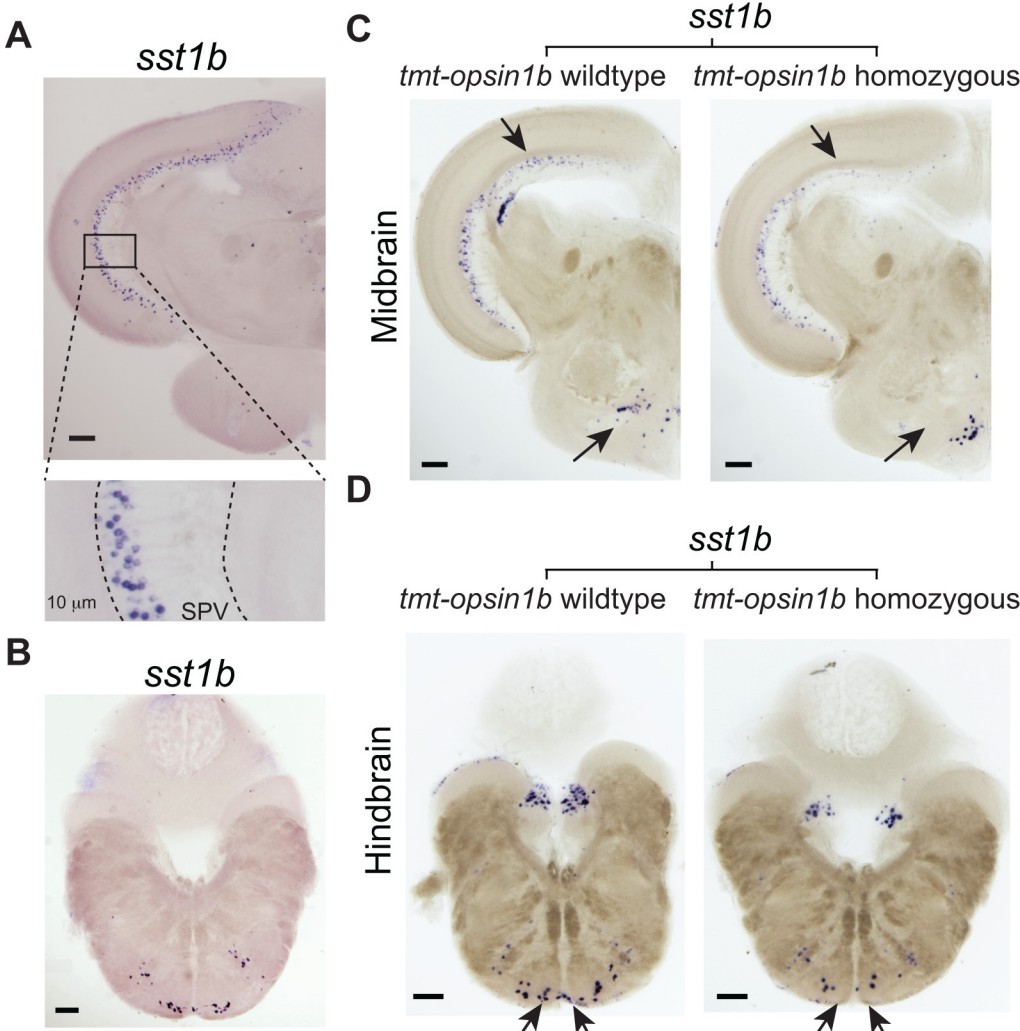

**Fig 6.** ***sst1b* levels are reduced in specific cells in the tectum, posterior hypothalamus, and hindbrain in *tmt-opsin1b*<sup>−/−</sup> medaka brains.** In situ hybridization for *sst1b* performed on coronal sections of midbrain (A) and hindbrain (B) from Cab wild-type fish. Magnification of boxed areas of the optic tectum are displayed below. SPV: stratum periventriculare. Note that *sst1b* expression identifies a distinct tectal population of dorsal cells and the ventral area of the hindbrain. (C, D) Comparison of *sst1b* expression in wild-type versus *tmt-opsin1b* mutant sibling coronal brain sections reveals specific reduction (denoted by arrows) in interneurons of the tectum and reticular formation, and in the posterior hypothalamus. Scale bars: 100 μm. Further details on anatomical annotations and co-localization of *sst1b* with *gad2*, as well as Scn12aa orthology relationships and localization: S6–S9 Figs.

separate from the *tmt-opsin1b*<sup>+</sup> cells (Figs 6B and 4D; anatomical annotation: [14], S6 Fig). We thus conclude, that the changes of *sst1b* transcript levels are rather indirectly mediated by *tmt-opsin1b*<sup>+</sup> cells upstream of *sst1b*-expressing cells.

We next analyzed where the regulation of *sst1b* transcripts occurs. We performed in situ hybridizations on wild-type and *tmt-opsin1b* mutant adult fish brains for *sst1b*, with specific attention to treating wild-type and mutant samples identically. We observed a specific reduction of *sst1b*<sup>+</sup> transcript levels in the cells of the stratum periventriculare (SPV) layer in the optic tectum, in the dorsal periventricular hypothalamic zone surrounding the lateral recess, and in cells medial to the nucleus glomerulosus in the medial preglomerular nucleus (arrows, Fig 6C). We also observed a reduction of *sst1b*<sup>+</sup> cells in the intermediate reticular formation of

the hindbrain (arrows, Fig 6D). In order to independently verify the specificity of the effects, we cloned *gad2*, whose ortholog is co-expressed with the zebrafish *sst1b* ortholog in neurons, based on single-cell RNA (scRNA) analyses [56]. We confirmed co-expression of *sst1b* with *gad2* in medaka tectal neurons (S7 Fig), and also did not find any visible alteration in the expression of *gad2* in *tmt-opsin1b* mutant versus wild-type brains (S7 Fig). Taken together, these in situ expression analyses confirm a likely specific effect of *tmt-opsin1b* on *sst1b* transcripts. They further suggest an indirect regulation of *sst1b* transcripts by *tmt-opsin1b*. We additionally identified that the tectal *sst1b*+ cells are GABA-ergic (by the expression of *gad2*), and hence likely convey inhibitory input to the tectal circuitry.

We likewise further analyzed *scn12aa* brain expression. Medaka Scn12aa is the common ortholog of the 2 separate amniote Scn10a and Scn5a groups (S8 Fig; S9 Data). Expression analyses of both mRNA (by in situ hybridization; S9A and S9B Fig) and antibody staining (using a commercially available antibody with broad species reactivity generated against human Scn5a; S9C–S9H Fig) suggest a broad, possibly ubiquitous brain expression. A specific set of neurons regulated by *tmt-opsin2* was not obviously detectable. It is hence plausible that the effect of *tmt-opsin2* is present in multiple neuronal populations and not restricted to a few specific circuits.

## Photoperiod regulates *sst1b* transcript levels via a *tmt-opsin1b*-dependent mechanism

We finally aimed to further investigate the possible light dependency of the observed transcript regulation. Given that *sst1b* exhibits the more specific expression, we focused on this transcript. Specifically, we tested whether changes in the illumination regime impact on *sst1b* transcript levels in a TMT-Opsin1b-dependent manner. We chose a "photoperiod"-type light regime for 3 reasons: First, any changes in transcription need sufficient time to occur; second, we wanted to test a light regime that has obvious natural relevance to medaka [30]; and, third, *pre-pro-somatostatin* expression is regulated in response to exposure to short- and long-day photoperiods in adult rats [7].

We exposed wild-type fish to 2 different white light regimes (Fig 7A and 7B). All fish were initially raised under a 16 h (light)/8 h (dark) white light regime (Fig 7B). One cohort remained exposed to this 16 h/8 h light/dark cycle (long day), while the other group was transferred to an 8 h/16 h light/dark cycle (short day). After 1 wk, eyes and brains were dissected at ZT 8 (long day) and ZT 4 (short day) (blue arrowhead, Fig 7B) and analyzed by qPCR.

As *sst1b* is circadianly regulated, with no phase difference between *tmt-opsin1b* mutants and wild-type fish (Fig 7C; S6 Data), we timed our brain part sampling to the middle of the respective light phases, an established sample timing strategy for photoperiod analyses in medaka fish [57].

A comparison of *sst1b* transcript levels at long- versus short-day light regimes revealed significantly lower transcript level for the short-day cohort, mimicking the *tmt-opsin1b* mutation (compare Fig 7D versus Fig 7E [S6 Data] and red versus black boxes in Fig 5B and 5C [S5 Data]). Comparing the transcript levels of *sst1b* under long-day versus short-day photoperiods in the *tmt-opsin1b* mutants revealed that this mutation leads to even stronger changes in *sst1b* transcript levels (Fig 7E; S6 Data). This is most apparent for the forebrain, where there are no changes between long- and short-day conditions in wild-type fish (Fig 7D; S6 Data), but a strong downregulation of *sst1b* under the short-day regime in the mutant (Fig 7E; S6 Data). This shows that the regulation of *sst1b* by light depends on *tmt-opsin1b*, but—similar to the behavioral changes in the avoidance assay—suggests that *tmt-opsin1b* normally prevents specific neurophysiology changes occurring in response to environmental light changes, a "stabilizing function" that is perturbed in *tmt-opsin1b* mutants.

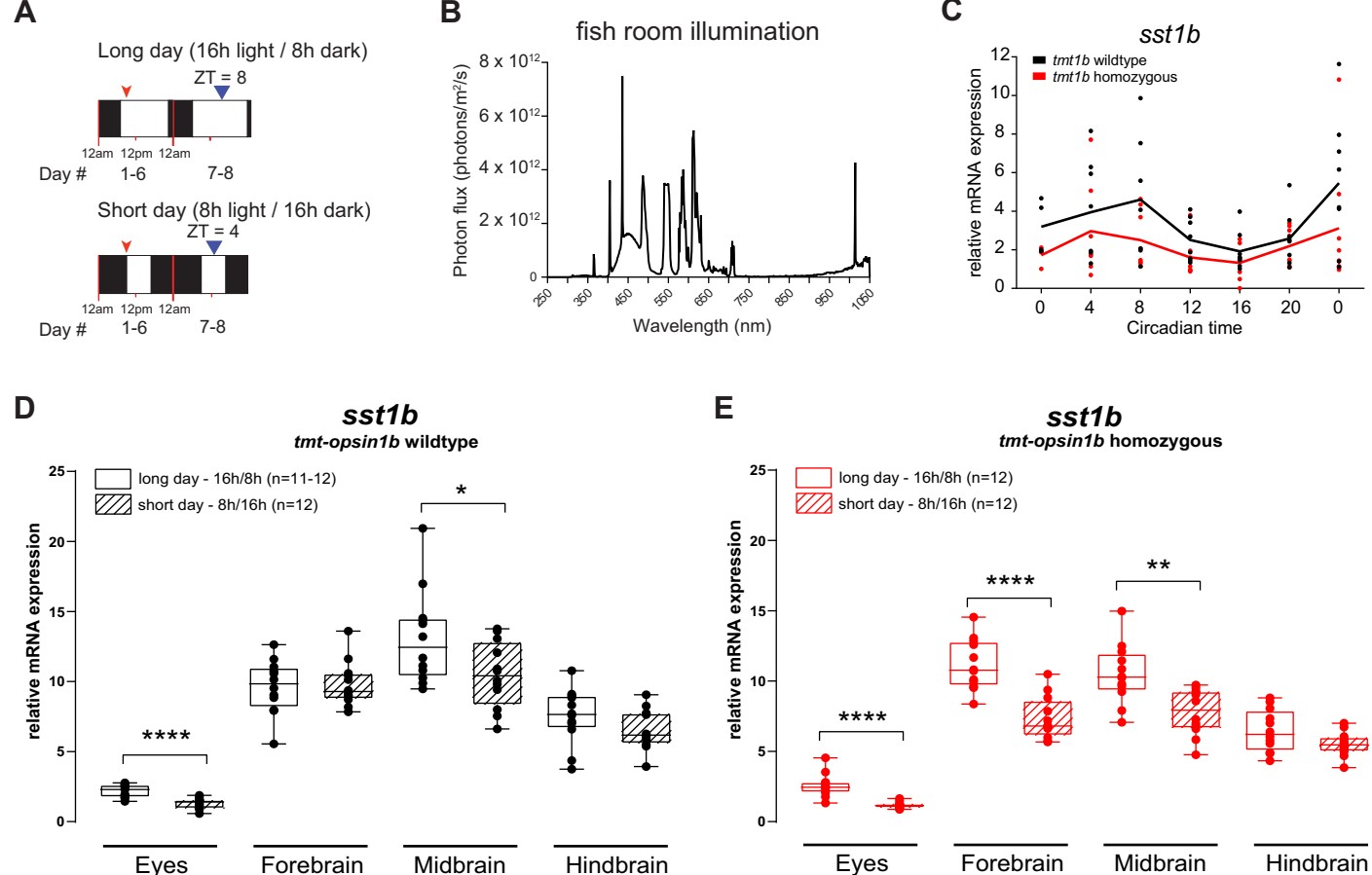

**Fig 7. *tmt-opsin1b* sensitizes *sst1b* regulation to changes in photoperiod.** (A) Schematic representation of the 2 different photoperiods assigned to different cohorts of *tmt-opsin1b* wild-type and *tmt-opsin1b* mutant adult fish. Black boxes: dark period; white boxes: white light period. Red arrowheads: feeding time; blue arrowheads: time when fish were sacrificed and the brain dissections made. (B) Light spectrum of the light used in (A). (C) Relative mRNA expression of *sst1b* in *tmt-opsin1b (tmt1b)* wild-type and mutant larvae sampled during constant darkness after entraining with 16 h/8 h white light/dark cycle at 27°C. (D, E) *sst1b* mRNA transcript expression levels in *tmt-opsin1b* wild-type (D) and *tmt-opsin1b* homozygous mutant (E) relatives. $^{****}P \leq 0.0001$, $^{**}P \leq 0.01$, $^{*}P \leq 0.05$. Box plots: single bar: median; box: first and third quartile; whiskers: minimum and maximum. Each data point represents 1 biological replicate. Metadata: S6 Data. For further details on spectra: S3 Fig.

## Discussion

The large number of *opsin* genes uncovered over the past years in vertebrates, particularly in teleosts, is puzzling, even more since many of these *opsin* genes have been shown to encode functional light receptors and are expressed at various places beyond the rods and cones (e.g., [9–12,14–17]). Here we have started to functionally tackle this diversity by investigating loss-of-function mutants of 2 nonvisual opsins of the ETO family in medaka.

The results are multifaceted and complex, and provide food for thought from multiple perspectives.

While first focusing on *tmt-opsin1b*, we showed that this opsin modulates fish swimming behavior in a light-dependent manner in several assays. Interestingly, in most cases *tmt-opsin1b* mutant fish display enhanced behavioral responses. At present we can only speculate about the contribution of the different *tmt-opsin1b*+ neurons in the response circuitry. While the contribution of several *tmt-opsin1b*+ cells is unclear, it is highly likely that tectal cells are involved. This may suggest the following circuitry: *tmt-opsin1b*+ tectal cells are positive for

ChAT [14]. These cells connect—possibly monosynaptically (given the close vicinity and branching of ChAT tectal neurons—see [14])—to neurons marked by *sst1b/gad2*. We showed that TMT-Opsin1b affects the production of *sst1b*. A member of the somatostatin/corticostatin family in the mouse (SST, active form SST-14) has been shown to itself suppress excitatory inputs to downstream interneurons, resulting in enhanced visual gain and orientation selectivity [58]. It might hence be plausible to assume that tectal *sst1b* also exerts suppressive effects. In *tmt-opsin1b* mutants, *sst1b* is downregulated in tectal interneurons, which would reduce such a suppressive effect and result in enhanced behavioral responses. However, we showed that these *sst1b*[+] neurons also express *gad2*. It might thus also be imaginable that GABA release could be modulated by TMT-Opsin1 in these neurons. Thus, focusing on the tectal circuitry, there are at least 2 possibilities for how loss of *tmt-opsin1b* results in increased activity. However, future work on specific neuronal circuits will be required to really understand the role of the different *tmt-opsin1b*-expressing neurons.

We also showed that the response differences present in *tmt-opsin1b* mutant versus wild-type fish, as well as the effects of the *tmt-opsin1b/tmt-opsin2* double mutation, are strongly age-dependent. As the expression domains of *tmt-opsin1b* and *tmt-opsin2* do not observably change between young larvae and adults [14], we think that the most plausible explanation for the age dependency is the still ongoing differentiation of the nervous system during young larval stages. Ontogenetic changes in behavior are very common, and well documented for visual processing and social behavior in medaka fish [53,59]. While the cellular and molecular causes still remain to be determined, it is clear that the phenomenon is common among vertebrates and even extends to humans across the lifespan, where it has been shown to correlate with changes in glutamatergic and GABAergic systems [60,61]. Especially the latter is interesting in the context of our analyses, as the *sst1b*[+] neurons downstream of *tmt-opsin1b* function in the tectum are GABAergic and might hence be subject to ontogenetic changes. Understanding behavioral changes across the lifespan is not just an important topic to better comprehend the extent of natural behavioral plasticity. It is also relevant to understanding behavioral adaptations to natural environments, as ontogenetic changes are often intertwined with habitat shifts, to balance the different risks and needs of individuals at different ages, which are often correlated with size differences [62,63].

Another level of molecular modulation we uncover is the regulation of *sst1b* by photoperiod. Short-day photoperiods result in levels of *sst1b* in mid- and hindbrain that are equal to those of the *tmt-opsin1b* mutants. However, in the mutants these levels become reduced even further, and *sst1b* reduction also occurs strongly in the forebrain, where it is unaffected in wild-type fish under long- versus short-day regimes. These data suggest that *tmt-opsin1b* normally prevents molecular alterations happening when environmental conditions change.

Interestingly, different members of the *somatostatin/corticostatin* family are frequently connected to changes in light parameters. A change of the preprohormone transcript and peptide levels of Somatostatin1 has been reported for long versus short photoperiods in adult rats [7], suggesting a possible evolutionary conservation of this regulatory relationship. Similar to the changes we observed for *sst1b* in medaka, SST1 amounts increase when the days are longer. In the case of the rat, this is correlated with a partial decrease of the enzyme tyrosine hydroxylase and the number of dopaminergic neurons, as well as an alteration in stress behavior [7]. However, in addition to the changes in the forebrain (posterior hypothalamus), as they were reported for rat, we find *sst1b* to be significantly regulated in mid- and hindbrain cells. It could be that this also extends to rat, and these findings certainly suggest that brain regions outside the hypothalamus should be analyzed, in order to understand which effects photoperiod differences might exert on mammals. It will then be for future research to sort out which molecular changes are related to which behavioral changes.

*sst1.1* in zebrafish has been implicated in the regulation of behavioral search strategies upon sudden loss of illumination, a behavior that also depends on the melanopsin-type light receptor *opn4a*. However, it remains to be clarified if *sst1.1* functions downstream of or in parallel pathways to *opn4a*. It might be that different nonvisual opsins relay different light information via different *somatostatin/corticostatin* members.

Finally, one of our main aims was also to obtain first insight into the functional interaction of different photoreceptors in the organism. Unexpectedly for us, both on molecular and behavioral levels, the simultaneous loss of 2 *tmt-opsin* genes can compensate for the loss of each of them individually, suggesting that they act at least in part in opposite directions on the same downstream circuitry. In this respect the expression of *tmt-opsin1b*, *tmt-opsin2*, and *sst1b* in the tectal region is again particularly interesting. *tmt-opsin1b* and *tmt-opsin2* tectal expression is non-overlapping, delimiting 2 distinct cellular populations, within 2 unique layers, while *sst1b* either overlaps or is directly adjacent to *tmt-opsin2*[+] cells. This spatial expression raises the possibility that both *tmt-opsin* genes (and *sst1b*) might together contribute to the local neuronal circuit outlined above. Future work is needed not only to further delineate these interactions, but also to better understand their significant complexity in the context of natural light habitats and adaptations.

## Materials and methods

### Medaka fish rearing and ethics statement

All animal research and husbandry were conducted according to Austrian and European guidelines for animal research (fish maintenance and care approved under BMWFW-66.006/0012-WF/II/3b/2014, experiments approved under BMWFW-66.006/0003-WF/V/3b/2016, which is cross-checked by Geschäftsstelle der Kommission für Tierversuchsangelegenheiten gemäß § 36 TVG 2012 p. A. Veterinärmedizinische Universität Wien, A-1210 Wien, Austria, before being issued by the Bundesministerium für Bildung, Wissenschaft und Forschung). Medaka fish (*O. latipes*) strains were kept in a constant recirculating system at approximately 26–28˚C in a 16 h light/8 h dark cycle and bred using standard protocols [33]. Collected embryos were kept at 28˚C until hatching in Embryo Rearing Medium (ERM; 0.1% [w/v] NaCl, 0.003% [w/v] KCl, 0.004% [w/v] $CaCl_2 \times H_2O$, 0.016% [w/v] $MgCl_2 \times 6 H_2O$, 0.017 mM HEPES, and 0.0001% [w/v] methylene blue). Mutant lines were generated in and outcrossed at least 4 times to the Cab wild-type background (*tmt-opsin1b* Δ+23 was outcrossed 13 times). Where available, different mutant alleles were used interchangeably, thereby reducing the probability of effects induced by off-target mutations. Mutant versus sibling wild-type batches and batches from heterozygous crosses were positioned in random, alternating positions relative to the respective light sources while grown prior to the assays. Parental fish grew in as similar fish room light conditions as possible.

### Design and construction of *tmt*-TALENs

TALENs targeting the first exon of medaka *tmt-opsin1b* (Ensembl gene ENSORLG00000013181) and *tmt-opsin2* (Ensembl gene ENSORLG00000012534) were designed using TAL Effector Nucleotide Targeter v2.0 (https://tale-nt.cac.cornell.edu/) with the following parameters: equal left and right repeat variable diresidue (RVD) lengths, spacer length of 15–20 bp, NN for G recognition, only T in the upstream base, and the presence of a unique restriction site in the spacer region. TAL effector modules were assembled and cloned into the array plasmid pFUS using the Golden Gate TALEN and TAL Effector Kit (Addgene, Cambridge, MA, US) according to validated procedure [64]. The RVD sequence of the left *tmt-opsin1b*-TALEN was HD NN NG NG HD HD HD NI NI HD NN HD NN NI NN HD

HD NG, binding to the genomic sequence 5′-CGTTCCCAACGCGAGCCT-3′, and the RVD sequence of the right *tmt-opsin1b*-TALEN was NN HD NG HD HD NN HD NG NN HD NN NG HD HD HD HD NN NG, binding to the genomic sequence 5′-GCTCCGCTGCGTCC CCGT-3′ on the opposite strand, featuring a unique BssHII restriction site in the 17-bp spacer region. The RVD sequence of the left *tmt-opsin2*-TALEN was NN NG NG NG NG HD HD NN NN NG HD NI NN NI HD, binding to the genomic sequence 5′-GTTTTCCGGTCA GAC-3′, and the RVD sequence of the right *tmt-opsin2*-TALEN was NI NG HD NN HD NG HD NG NN NN NG NG NN NI, binding to the 5′-CATCGCTCTGGTTGA-3′ genomic sequence. The final pCS2+ backbone vectors, containing the homodimeric FokI nuclease domains, were used as previously [65]. All final TALEN pCS2+ expression vectors were sequence-verified using 5′-TTGGCGTCGGCAAACAGTGG-3′ forward and 5′-GGCGAC GAGGTGGTCGTTGG-3′ reverse primers.

## Generation of capped TALEN mRNA and microinjection

Full-length TALEN plasmids were digested with KpnI, gel purified (Gel Extraction Kit, Qiagen, Venlo, Netherlands), and in vitro transcribed (Sp6 mMESSAGE mMACHINE kit, Thermo Fisher Scientific, Waltham MA, US), followed by a purification using the RNeasy Mini Kit (Qiagen). The yield was estimated by NanoDrop (Thermo Fisher Scientific) and diluted to the final concentration for microinjection. Cab strain zygotes were microinjected with a mix containing 5 or 50 ng/μl of each transcribed TALEN mRNA and 0.6% tetramethylr-hodamine (TRITC; Thermo Fisher Scientific) in nuclease-free water.

## Genotyping of *tmt-opsin1b*- and *tmt-opsin2*-TALEN mutated fish

Genomic DNA from hatched larvae or caudal fin biopsies were retrieved with lysis buffer (0.1% SDS; 100 mM Tris/HCl [pH 8.5]; 5 mM EDTA; 200 mM NaCl; 0.01 mg/ml Proteinase K) at 60°C overnight, and 1 μl of a 1:20 dilution of the lysate was used in a PCR reaction (Hot-StarTaq, Qiagen). The sequence flanking the *tmt-opsin1b* TALEN binding sites was amplified using the forward 5′-GGGACTTTCTTTGCGCTTTA-3′ and the reverse 5′-CAGGTCAGA GCGGATCTCAT-3′ primers. Ten microliters of the reaction was directly used for restriction digest using 4 units of BssHII enzyme (New England Biolabs, Ipswich, MA, US) in a 20-μl reaction. Genotyping of the *tmt-opsin2* locus was made with the forward 5′-CGGTGAGC GATGTGACTG-3′ and the reverse 5′-GGGAGATCTTTGTCCAGGTG-3′ primers. Mutation caused by TALENs was assessed by analyzing the band sizes of the restriction digest product on a 2% agarose gel. Undigested bands were gel extracted and subcloned into pJET2.1 using the Clone JET PCR Cloning Kit (Thermo Fisher Scientific) and sequence-verified.

## Immunohistochemistry

**Without clearing.** Adult fish (>2 months old) were anesthetized in fish water containing 0.2% tricaine and decapitated; the brain was dissected, fixed in ice-cold 4% PFA overnight, and subsequently stored in 100% methanol for at least 1 wk before use. After rehydration in successive steps of PBT (PBS with 1% Triton X-100), the dissected brains were fixed in 4% paraformaldehyde in PBT for 2 h, embedded in 3% agarose, and cut into slices of 100 μm thickness on a tissue vibratome, followed by 4 h of blocking in 5% sheep serum at room temperature (RT). Slices were then incubated for 3 d at 4°C with a polyclonal rabbit anti-Ola-TMT-1b antibody (1:250 in 1% sheep serum). After extensive PBT washes, sections were treated with the secondary antibody goat anti-rabbit Alexa Fluor 488 (Thermo Fisher Scientific) in a 1:500 dilution in 1% sheep serum supplemented with 1:10,000 of 4′,6-diamidino-2-phenylindole (DAPI), during 2 d at 4°C. Following PBT washes, slices were mounted in

proper mounting medium, and pictures were taken on a Zeiss LSM700 confocal scanning microscope.

**With clearing.** The protocol was adapted from [66] with the following details: Adult fish (>2 months old) were anesthetized in fish water containing 0.2% tricaine and decapitated; the brain was dissected and fixed in ice-cold 4% PFA overnight. After washes in 1× PBS at RT, the brains were incubated in pre-chilled acetone O/N at −20˚C, rehydrated in 1× PBS at RT, and digested with 10 μg/ml Proteinase K (Merck #1245680100) for 10–12 min at RT, followed by 2× glycine washes (2 mg/ml) (Roth #3908.3). Brains were washed in 1× PBS, immersed in Solution 1.1 (3 h, at 37˚C) under gentle shaking, subsequently washed in 1× PBS O/N, embedded in 3% agarose, and sliced (100-μm thickness) using a vibrating blade microtome (Leica VT1000S, Leica Biosystems, Germany). The slices were then blocked for 1–2 h in 10% sheep serum/1× PBS, followed by primary antibody incubation in 10% sheep serum at 4˚C for 3 d (anti-SCN5A antibody produced in rabbit; Sigma SAB2107930; 1:1,000). After extensive 1× PBS washes, sections were incubated in a goat anti-rabbit Cy3 antibody (Thermo Fisher Scientific; 1:500 dilution) in 10% sheep serum, 1:10,000 DAPI was added, and the sections were incubated for 3 d at 4˚C. Following PBS washes, slices were mounted in proper mounting medium, and pictures were taken on a Zeiss LSM700 confocal scanning microscope.

## Avoidance assay

The behavioral assay was essentially performed as described before [40]. A single medaka larva (9–12 dpf; unfed) or juvenile fish (20–22 dpf; fed) was placed in a custom-built acrylic glass chamber (18 × 14 × 305 mm) with a transparent bottom and opaque side walls and filled with 1× ERM (1 cm high). Five chambers lying parallel to each other were placed on a horizontal-facing computer screen, allowing for the simultaneous recording of 5 free-swimming animals in the same behavioral trial. Stimuli consisted of moving black dots (except when testing for contrast, where different abstract gray values were used) on a white background traveling at a constant speed of 13.5 mm/s in the same direction in every trial. Each behavior trial consisted of 7 blocks of 6 dots each, with each block characterized by a unique dot size, presented in a size-ascending or pseudo-random order. Stimuli were generated on the computer screen using custom-written programs (Ubuntu). Two consecutive blocks were presented with a 19-s interval in between during which no stimulus was displayed. The size of the dot was characterized by its diameter (in pixels) and was calculated as degrees of the larva visual field, as described before [40]. An industrial camera (acA1300-30gm GigE; Basler, Germany), positioned 40 cm above the computer screen, allowed visualization of the fish–dot interactions, recording videos at a frame rate of 33 frames per second for offline analysis. The fish were allowed to accommodate to the chamber for 5 min before the beginning of the trial, which is longer than in previous similar assays in zebrafish [40] and *Xenopus* [37], likely ensuring a robust standardization across different experiments. The experiments were performed during the natural light part of the fish light/dark cycle, between ZT 2 and ZT 5, and a single animal was used in only 1 behavior trial to avoid possible habituation (e.g., [67]) or learning. Each fish–dot interaction was scored either as an approach, when the fish swam toward a moving dot, or an avoidance, when the fish swam away from a moving dot [40]. A neutral interaction was scored when the dot entering the fish visual field caused no change in the initial swimming pattern of the animal or if the fish remained motionless. Approach behaviors consisted of swimming towards the dot, displaying a distinct attraction and predation behavior [67], whereas avoidance behaviors were faster than approaches, characterized by rapid swimming bouts away from the direction of the dot's movement. All video analyses were made prior to genotyping of the fish to shield the identity of the subjects from the observer. An AI ([number

avoidances − number approaches]/[total number of dots in a block]) was used as a quantitative behavioral readout. All behavioral assays were made at a constant RT of 26˚C, with the room lights switched off and the behavioral setup covered by a black, light-impenetrable cover cloth, this way ensuring that the computer screen would be the only illumination source. The Michelson contrast [68] ($[L_{max} − L_{min}]/[L_{max} + L_{min}]$) was used to quantify the relative difference in luminance between the moving dot and the computer screen background. $L_{max}$ and $L_{min}$ are luminance maximum and minimum, respectively. Medaka are known to have highly acute vision, as demonstrated by previous studies on the optomotor response [53,69], showing excellent visual performance already from hatching [69].

## Photokinesis assay

Individual larvae (9 to 12 dpf; unfed; kept at a 16 h/8 h light cycle at 28˚C until hatching) or juvenile medaka fish (20 to 22 dpf; fed) were distributed randomly across a 6-well plate containing 10 ml of 1× ERM per well. A behavioral trial consisted of a behavioral paradigm that evaluates the animal swimming activity (i.e., distance moved) during alternating blocks of 30 min of light and darkness (3 h in total per trial) [70], at a constant temperature of 27˚C. An initial acclimation phase of 5 min to darkness was used before the start of the trial. When assessing the swimming responses in a more naturalistic assay, a photoperiod of 16 h light/8 h dark was used during 2 consecutive days, at a constant temperature of 27˚C. Behavioral assessment was made between ZT 3 and ZT 12. DanioVision (Noldus, The Netherlands) hardware was used to track the distance moved by the animals in each trial. Larval motion was tracked at 60 frames/s over the trial. In an alternate experiment, 1-min pulses of white light and darkness were presented to the larvae, during a total trial duration of 7 min (S2A Fig). Video data were posteriorly analyzed offline by the tracking software EthoVision XT (Noldus) to calculate the average distance moved by the animal every 10 s of the trial. When comparing the dark photokinesis between *tmt-opsin1b* mutant and wild-type siblings (S2M and S2N Fig), we normalized the different baseline levels by taking the average distance moved during the 5 min of preceding darkness in the trial as a normalization factor. Each animal was used only once per trial, thus avoiding possible habituation biases. An ILT950 spectrometer (International Light Technologies, US) was used to measure the spectra and intensity of the different light sources. A light-impenetrable cloth was placed over the behavior setup, thus assuring a lack of external light contamination.

## Diel swimming test

Care was taken to start each trial at the precise light-to-dark transition, thus minimizing any deleterious effect that the shift to a new environment might impose on the natural behavior of the fish. Since scheduled food availability has an impact on the locomotor activity of fish [71], animals were kept unfed during the 2 consecutive days of the experiment, thus ensuring that the change in ambient light was the central feature impacting on swimming responses. The temperature was kept constant at 27˚C. The setup, tracking, and off-line analysis were done as previously described for the "photokinesis assays," with the exception that the activity of the animals was averaged every 1 min.

## Enucleations

Enucleations were performed on 13-dpf wild-type and *tmt-opsin1b* mutant juvenile fish upon anaesthetization with 0.03% tricaine (MS-222) (Sigma, St. Louis, MO, US). After the surgery, fish were allowed to recover overnight in fresh rearing fish water at 26˚C. At 1 wk post-surgery, the fish were used for behavior testing. Handling control fish were anaesthetized for the same duration and placed afterwards in fresh fish water overnight.

## RNA extraction and sequencing

For RNA-seq experimentation, the eyes, forebrain, midbrain, and hindbrain (anatomical boundaries according to [72]) were dissected from age-matched wild-type and mutant *tmt-opsin1b* and *tmt-opsin2* adult (>2 months old) medaka sibling fish. One metal bead (Peqlab Biotechnologie, Germany) and 350 μl of RLT buffer (Qiagen) including 1% β-mercaptoethanol was added to the frozen tissue parts, followed by homogenization for 3 min at 30 Hz using the Qiagen tissue lyser. Total RNA was extracted using the RNeasy Mini Kit (Qiagen) according to manufacturer's protocol. Three independent biological replicates, each made of 3 individual fish (total of 9 fish per genotype) were used. For *tmt-opsin1b*, 1 fish per independent mutation was used. The quality of total RNA was checked using the Agilent RNA 6000 Nano Kit (Agilent, US) and then enriched for poly(A)+ RNA using the Dynabeads mRNA Purification Kit (Thermo Fisher Scientific). We used the SuperScript VILO cDNA Synthesis Kit (Thermo Fisher Scientific) to generate strand-specific cDNA that was further sequenced on the Illumina HiSeq 2500 platform by the Vienna Biocenter Core Facilities NGS Unit (http://www.vbcf.ac.at) as 100-base single-end reads, resulting in 15–61 million reads, on average, per biological replicate.

## Differential gene expression

Sequences from each sample were mapped against the assembled chromosomes of the medaka genome (Ensembl version 96) using the read mapper NextGenMap [73]. After filtering for duplicates and low quality base, the stand-alone *featureCounts* tool [74] was used to count the mapped reads for each transcript in each sample. The Bioconductor package edgeR (version 3.9) was used to analyze read count data and to identify differential expression levels (Benjamini–Hochberg method). We classified genes as being differentially expressed between genotypes when the differences in expression level between wild-type and homozygous mutant fish were significant at a false discovery rate (FDR) of 5%.

## qPCR

One metal bead (Peqlab Biotechnologie) and 700 μl of TRI Reagent (Sigma-Aldrich, St. Louis, MO, US) were added to the previously dissected and frozen eyes, forebrain, midbrain, and hindbrain, followed by homogenization for 2 min at 30 Hz using the Qiagen tissue lyser. Total RNA was extracted from the different brain parts and eyes using Direct-zol RNA MiniPrep (Zymo Research, US) according to the manufacturer's protocol. Then 80–200 ng of total RNA was transcribed into cDNA using the QuantiTect Reverse Transcription Kit (Qiagen) with random hexamer primers. Each cDNA was further analyzed in duplicate in a 25-μl volume using a SYBR-green-containing mastermix in the Applied Biosystems (Thermo Fisher Scientific) mastercycler. Intron-spanning qPCR primers were designed with the Universal ProbeLibrary software from Roche (Table 1). Expression of each gene was normalized to *beta-actin* transcript levels, and fold changes were calculated.

## Cloning of medaka *gad2* and *scn12aa*

A BLAST search using zebrafish *gad2* identified predicted homologous sequences; homology was confirmed by reciprocal back-blast (for *gad2*).

cDNAs were cloned following standard protocols, using Phusion DNA Polymerase (NEB) and cloned in pJET1.2. Inserts were verified by sequencing.

Primers (5′→3′) were as follows: *gad2*, forward atggcatctcacgggttctggtctc, reverse ttatagat catggccgaggcgttca; *scn12aa*, forward cgccatgcagagagaagattag, reverse gacactcggtggtgaagattac. GenBank accession numbers are MT267535, MT239386, and MT793601.

**Table 1. Forward and reverse primer sequences used for quantitative PCR.**

| Gene | Forward primer (5′→3′) | Reverse primer (5′→3′) | Ensembl Gene ID |
|---|---|---|---|
| *sst1b* (also named *sst3* or *cort*) | ggcttcctgtggaggaca | cagacaccagcttaaggatca | ENSORLG00000027736 |
| *beta-actin* | gtgctgtctttccctccatc | tagctgtctttctggcccat | ENSORLG00000013676 |
| *scn12aa* | gcagatgacctgtcggaact | ctgacagtgcctcagacagaa | ENSORLG00000011677 |
| *per1b* | ggggagagtcgtgtacgtgt | tgctgctgtagaaggtgctg | ENSORLG00000019370 |
| *reverb*β | aggagcatccagcagaacat | ctggctgtttttgctcatca | ENSORLG00000016431 |

## In situ hybridization (ISH) on adult brain sections

The generation of antisense digoxigenin (DIG)–labeled RNA probes and the ISH staining procedure on adult brain sections were performed as previously [14]. Briefly, adult fish (>2 months old) were anesthetized in fish water containing 0.2% tricaine and decapitated, and the brain was dissected, fixed in ice-cold 4% PFA overnight, and subsequently stored in 100% methanol for at least 1 wk prior to use. After rehydration in 1× PTW (PBS + 0.1% Tween-20), brains were digested with fresh 10 µg/ml Proteinase K (Merck, US) for 35 min, followed by a 4-h prehybridization step at 65˚C with Hyb+ solution (50% formamide, 5× SSC [pH 6.0], 0.1% Tween-20, 0.5 mg/ml torula [yeast] RNA, 50 µg/ml heparin). Incubation with the specific DIG-RNA probe was made overnight at 65˚C. Coronal whole-brain slices (100-µm thickness) were made using a vibrating blade microtome (Leica VT1000S, Leica Biosystems, Germany), blocked for >1 h in 10% sheep serum/1× PTW, followed by incubation with a 1:2,000 dilution of anti-DIG-alkaline-phosphatase-coupled antibody (Roche, Switzerland) in 10% sheep serum overnight. Detection of DIG probes was made in staining buffer (in 10% polyvinyl alcohol) supplemented with nitro-blue tetrazolium (NBT) and 5-bromo-4-chloro-3′-indolyphosphate (BCIP) (Roche). In addition, the *gad2* probe was co-detected fluorescently as described in [75].

## Cortisol assay

**Mechanical disturbance.** Larvae were placed onto an orbital shaker (300 rpm) for 5 min, subsequently sacrificed, and snap-frozen in liquid nitrogen.

**Cortisol extraction and measurement.** The procedure for extracting cortisol from whole larvae was adapted from [76]. Briefly, the larvae were sacrificed and frozen in liquid nitrogen, and stored at −80˚C until further processed. The samples were thawed on ice, and 200 µl of sterile PBS was added. The samples were subsequently homogenized at RT for 1 min at 30 Hz (Tissue Lyser II, Qiagen). Twenty microliters of each homogenate was removed for protein concentration measurements (Pierce BCA Protein Assay Kit, Thermo Fisher Scientific).

To the remaining 180 µl, 1.4 ml of ethyl acetate was added. The samples were subsequently centrifuged at 7,000*g* for 15 min, and the upper organic phase was collected within a glass vile. The samples were left to evaporate in a fume hood overnight. The dried samples were resuspended in 180 µl of ELISA buffer (as provided by the kit mentioned below).

The concentration of cortisol was measured using an enzyme-linked immunosorbent assay (ELISA) kit (500360, Cayman Chemical).

## Phylogenetic analysis

Sequences were aligned using the MUSCLE alignment algorithm (https://www.ebi.ac.uk/Tools/msa/muscle/). The resulting alignment was used to generate a maximum likelihood (ML) tree by the IQ-TREE web server [77], visualized using iTOl (https://itol.embl.de/) and exported as an EPS file. Coloration and text formatting were performed in Adobe Illustrator CC 2017. The accession numbers can be found in the respective metadata tables (S9 Data).

## Statistics

The D'Agostino and Pearson normality test was used to assess data distribution. The F-test for comparison of variances for normally distributed data was used to assess the variance of the standard deviation. When comparing 2 normally distributed groups of data with the same standard deviation, the unpaired *t* test was used. For comparing 2 groups of normally distributed data with different standard deviations, the unpaired *t* test with Welch correction was used. The Mann–Whitney test was used when comparing 2 non-normally distributed datasets. The software used was Prism version 8.

Data are presented as mean ± SEM. For all statistical tests used, a 2-tailed *P* value was chosen: $^{*}P \leq 0.05$; $^{**}P \leq 0.01$; $^{***}P \leq 0.001$; $^{****}P \leq 0.0001$.

## Supporting information

**S1 Data. Metadata for Fig 1.**
(XLSX)

**S2 Data. Metadata for Fig 2.**
(XLSX)

**S3 Data. Metadata for Fig 3.**
(XLSX)

**S4 Data. Metadata for Fig 4.**
(XLSX)

**S5 Data. Metadata for Fig 5.**
(XLSX)

**S6 Data. Metadata for Fig 7.**
(XLSX)

**S7 Data. Metadata for S1 Fig.**
(XLSX)

**S8 Data. Metadata for S4 Fig.**
(XLSX)

**S9 Data. Metadata for S8 Fig.**
(XLSX)

**S10 Data. Forebrain differentially expressed genes.**
(XLSX)

**S11 Data. Mid-/hindbrain differentially expressed genes.**
(XLSX)

**S1 Fig. Relates to Figs 1 and 4.** (A–F) Schematic of the predicted amino acid (aa) sequence for wild-type TMT-Opsin1b (A), wild-type TMT-Opsin2 (E), and corresponding recovered single TMT-Opsin1b (B–D) and TMT-Opsin2 (F) mutants. Blue aa: transmembrane domains; green aa: Lys296 forming the chromophore's Schiff base; red aa: substituted amino acids. (G) Partial *Ola-tmt-opsin1b* and *Ola-tmt-opsin2* genomic DNA sequence and corresponding aa codon translation for wild-type fish and corresponding mutants that were used interchangeably in the study (gray: TALEN binding site sequences; blue: recognition enzymatic site for BssHII; red: inserted nucleotides and substituted aa; asterisk: stop codon). All the *tmt-opsin1b* mutants had in common the disruption of the BssHII recognition site (used for mutagenesis PCR

confirmation) and a premature stop codon that led to a possible N-terminally truncated and hence null-functional protein. (H) Quantification of the number and identity of the wild-type and mutant allele siblings used in each trial. (I) Avoidance index levels of Cab wild-type larvae at different contrast luminance levels between the moving dot (constant 20˚ size) and the white computer screen background. Red arrowhead indicates the contrast level measured at 35% light intensity. Note that at this contrast value, the level of avoidance is comparable to the avoidance levels revealed at higher contrasts. (Metadata: S7 Data.)
(EPS)

**S2 Fig. Relates to Fig 1.** (A, B) Embryonic staging and development using morphological criteria for wild type and *tmt-opsin1b*$^{-/-}$ mutant according to [33]. Some representative details: Stage 10 shows for both wild type and mutant a thick blastoderm, with smaller inner cells. At Stage 19 both wild type and mutant show the appearance of otic vesicles (ov), as indicated. Furthermore, a groove in the optic lobes can be observed. With Stage 26, both wild type and mutant show that the choriodea start to cluster, as indicated by the arrow, and the guanophores (gp) are clearly visible. At Stage 30, the pectoral fin (pf) is apparent for both, as well as the otholits (ot). For Stage 36, the tip of the tail reaches the otic vesicle and the guanophores are now distributed from head to tail. Scale bar: 500 μm. (C) The table shows the total number of embryos analyzed, with representatives depicted in (A). Both wild type and mutant have an *n* = 20. (D–G) Representative images of wild-type fish and *tmt-opsin1b*$^{-/-}$ mutants shortly before hatching; GFP expression in the fish's retinal ganglion cells shows no noticeable difference in axonal projections. Scale bars: 100 μm.
(TIF)

**S3 Fig. Relates to Figs 1–4 and 7.** The units from our data are converted and plotted on a logarithmic scale to facilitate comparisons with the published dataset. (A) Underwater light measurements for individual wavelengths between August 24 and 25, 1999 [78]. For calculations considering water turbidity also see [79]. (B) Light spectra of the fish room lighting, in which the fish were raised if not stated otherwise. (C, D) Light spectra of Fig 1I. (E, F) Light spectra of S4A Fig. (G–I) Light spectra of Fig 2I, 2K and 2M plotted as irradiance.
(EPS)

**S4 Fig. Relates to Fig 2.** (A) Light spectra of the different LED lights used in the photokinesis assays (white light: yellow trace; blue light: blue trace). (B–I) Profile of average distance moved during alternating 30-min light/dark intervals and corresponding locomotor activity (as measured by the area under the curve) for 8 dpf (B, C), 9 dpf (D, E), 10 dpf (F, G), and 11 dpf (H, I) larvae. Each data point represents the mean (± SEM) distance moved for the preceding 10 s. (J, K) Dark photokinesis response normalized to the average of the 5 min of preceding darkness for larvae (J) and juvenile fish (K). All panels: black: *tmt-opsin1b* wild type; red: *tmt-opsin1b* mutant. Colored boxes along the *x*-axis represent light condition. Box plots: single bar: median; box: first and third quartile; whiskers: minimum and maximum. a.u.: arbitrary units. (Metadata: S8 Data.)
(EPS)

**S5 Fig. Relates to Figs 3 and 4.** (A–T) Individual and higher time-resolved plots of average distance moved in daytime shown for the first 1.5 h of the first (A–E and K–O) and second day (F–J and P–T) of the experiment in larvae (A–J) and juvenile fish (K–T). Each data point represents the mean (± SEM) distance moved for the preceding 1 min. Metadata for A, B, F, G, K, L, P, Q: S3 Data. Metadata for C, D, E, H, I, J, M, N, O, R, S, T: S4 Data.
(EPS)

**S6 Fig. Relates to Fig 6.** Anatomical annotation of *sst1b*$^+$ cells as revealed by in situ hybridization in the adult medaka fish brain, coronal/cross (A–H) and sagittal (I) sections.

Abbreviations: Hc, caudal hypothalamus; Hd, dorsal hypothalamus; PGZ, periventricular gray zone; Pr, pretectum; PPp, posterior parvocellular preoptic nucleus; SCN, suprachiasmatic nucleus; TH, tuberal hypothalamus; TPp, periventricular posterior tuberculum; Vd, dorsal nucleus of ventral telencephalic area; Vv, ventral nucleus of ventral telencephalic area; Vl, lateral nucleus of ventral telencephalic area; V, ventral telencephalic area.
(TIF)

**S7 Fig. Relates to Fig 6.** (A) *sst1* co-expression with *gad2* in tectal neurons (arrows). (B, C) Representative images of in situ hybridizations for *sst1b* and *gad2* on coronal sections of the midbrain from *tmt-opsin1b* wild-type (B) and *tmt-opsin1b* homozygous mutant (C) fish document no difference between wild-type and mutant fish. Red channel: *gad2*. Scale bars: 200 μm.
(TIF)

**S8 Fig. Relates to Fig 6.** Maximum likelihood (ML) tree to resolve the orthology relationships of different voltage-gated sodium channel subunits across vertebrates. Scn12aa from medaka fish is part of an ancestral fish/amphibian group that duplicated in amniotes, giving rise to 2 distinct groups, Scn5a and Scn10a. Variants of the *Drosophila* voltage-gated sodium channel PARA are used as outgroup. For accession numbers of the proteins used for alignment and tree construction, see S9 Data.
(EPS)

**S9 Fig. Relates to Fig 6.** (A, B) In situ hybridization for *scn12aa* performed on coronal sections of midbrain (A) and hindbrain (B) from Cab wild-type fish. Scale bar: 50 μm. (C–G) Confocal images of brains stained for Scn12aa, using an antibody raised against *Hsa*-Scn5a (red) and DAPI as nuclear counterstain (blue) on coronal sections of midbrain (C, D) and hindbrain (F, G) from Cab wild-type fish. White box: region corresponding to the 40× magnification. White arrowheads: representative positively stained neuronal projections and cell bodies. (E, H) Control slices with equal treatment as for (C, F), but no primary antibody added.
(TIF)

**S1 Video. Relates to Fig 1E.** Example of a 10-dpf *tmt-opsin1b* wild-type larvae performing the avoidance assay during the ascending size dot assay. Large dots elicit a distinct avoidance behavior.
(MP4)

**S2 Video. Relates to Fig 1E.** Example of a 10-dpf *tmt-opsin1b* single homozygous mutant larvae performing the avoidance assay during the ascending size dot assay. Large dots elicit a distinct avoidance behavior. Note that the displayed number of avoidances is higher when compared with wild-type larvae.
(MP4)

## Acknowledgments

We thank Andrij Belokurov and Margaryta Borisova for fish care; Andrew Straw and the members of the Tessmar-Raible, Raible, von Haeseler, and Baier labs for helpful discussions; Mario Wullimann for advice on medaka neuroanatomy; Mirta Resetar for preparation of fish brains and RNA for quantitative sequencing analyses; and Sven Schenk for sharing the mRNA sequencing protocol. Marin Čagelj contributed with the illustrations in Figs 1C, 4A, and 5A.

## Author Contributions

**Conceptualization:** Bruno M. Fontinha, Kristin Tessmar-Raible.

**Data curation:** Bruno M. Fontinha, Theresa Zekoll, Mariam Al-Rawi, Kristin Tessmar-Raible.

**Formal analysis:** Bruno M. Fontinha, Theresa Zekoll, Mariam Al-Rawi, Miguel Gallach, Florian Reithofer, Alison J. Barker, Ruth M. Fischer, Arndt von Haeseler, Herwig Baier.

**Funding acquisition:** Kristin Tessmar-Raible.

**Investigation:** Bruno M. Fontinha, Theresa Zekoll, Ruth M. Fischer.

**Methodology:** Bruno M. Fontinha, Mariam Al-Rawi, Maximilian Hofbauer, Ruth M. Fischer.

**Project administration:** Kristin Tessmar-Raible.

**Resources:** Kristin Tessmar-Raible.

**Supervision:** Kristin Tessmar-Raible.

**Validation:** Bruno M. Fontinha.

**Writing – original draft:** Bruno M. Fontinha, Kristin Tessmar-Raible.

**Writing – review & editing:** Bruno M. Fontinha, Theresa Zekoll, Mariam Al-Rawi, Miguel Gallach, Florian Reithofer, Alison J. Barker, Maximilian Hofbauer, Ruth M. Fischer, Arndt von Haeseler, Herwig Baier, Kristin Tessmar-Raible.

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
