## [Editor Report · Decision Letter 0]

4 Sep 2019

Dear Dr Tessmar-Raible, 

Thank you for submitting your manuscript entitled "TMT-Opsins differentially modulate medaka brain function in a context-dependent manner" for consideration as a Research Article by PLOS Biology.

Your manuscript has now been evaluated by the PLOS Biology editorial staff as well as by an academic editor with relevant expertise and I am writing to let you know that we would like to send your submission out for external peer review.

*Please be aware that, due to the voluntary nature of our reviewers and academic editors, manuscripts may be subject to delays during the holiday season. Thank you for your patience.*

Please re-submit your manuscript within two working days, i.e. by Sep 06 2019 11:59PM.

Kind regards,

Di Jiang,

Associate Editor

PLOS Biology

---

## [Decision Letter · Decision Letter 1]

5 Nov 2019

Dear Dr Tessmar-Raible,

Thank you very much for submitting your manuscript "TMT-Opsins differentially modulate medaka brain function in a context-dependent manner" for consideration as a Research Article at PLOS Biology. Your manuscript has been evaluated by the PLOS Biology editors, an Academic Editor with relevant expertise, and by three independent reviewers.

In light of the reviews (below), we will not be able to accept the current version of the manuscript, but we would welcome resubmission of a much-revised version that takes into account the reviewers' comments. If you have concerns or questions on how to best address the reviewers’ points, you are welcome to convey them in the form of revision plan, which will be passed on to the academic editor for advice. We cannot make any decision about publication until we have seen the revised manuscript and your response to the reviewers' comments. Your revised manuscript is also likely to be sent for further evaluation by the reviewers.

Your revisions should address the specific points made by each reviewer. Please submit a file detailing your responses to the editorial requests and a point-by-point response to all of the reviewers' comments that indicates the changes you have made to the manuscript. In addition to a clean copy of the manuscript, please upload a 'track-changes' version of your manuscript that specifies the edits made. This should be uploaded as a "Related" file type. You should also cite any additional relevant literature that has been published since the original submission and mention any additional citations in your response. 

Before you revise your manuscript, please review the following PLOS policy and formatting requirements checklist PDF: http://journals.plos.org/plosbiology/s/file?id=9411/plos-biology-formatting-checklist.pdf. It is helpful if you format your revision according to our requirements - should your paper subsequently be accepted, this will save time at the acceptance stage.

Please note that as a condition of publication PLOS' data policy (http://journals.plos.org/plosbiology/s/data-availability) requires that you make available all data used to draw the conclusions arrived at in your manuscript. If you have not already done so, you must include any data used in your manuscript either in appropriate repositories, within the body of the manuscript, or as supporting information (N.B. this includes any numerical values that were used to generate graphs, histograms etc.). For an example see here: http://www.plosbiology.org/article/info%3Adoi%2F10.1371%2Fjournal.pbio.1001908#s5.

For manuscripts submitted on or after 1st July 2019, we require the original, uncropped and minimally adjusted images supporting all blot and gel results reported in an article's figures or Supporting Information files. We will require these files before a manuscript can be accepted so please prepare them now, if you have not already uploaded them. Please carefully read our guidelines for how to prepare and upload this data: https://journals.plos.org/plosbiology/s/figures#loc-blot-and-gel-reporting-requirements.

Upon resubmission, the editors will assess your revision and if the editors and Academic Editor feel that the revised manuscript remains appropriate for the journal, we will send the manuscript for re-review. We aim to consult the same Academic Editor and reviewers for revised manuscripts but may consult others if needed.

We expect to receive your revised manuscript within two months. Please email us (plosbiology@plos.org) to discuss this if you have any questions or concerns, or would like to request an extension. At this stage, your manuscript remains formally under active consideration at our journal; please notify us by email if you do not wish to submit a revision and instead wish to pursue publication elsewhere, so that we may end consideration of the manuscript at PLOS Biology.

When you are ready to submit a revised version of your manuscript, please go to https://www.editorialmanager.com/pbiology/ and log in as an Author. Click the link labelled 'Submissions Needing Revision' where you will find your submission record. 

Sincerely,

Di Jiang, PhD

Associate Editor

PLOS Biology

Reviewer remarks:

Reviewer #1: This is a very clear and easy to follow paper by Fontinha and colleagues. Since there are ~40 opsins with likely redundant and even, as evident here, opposite roles, the idea to test the effect of individual opsins is quite heroic, and it not surprising that most of the effect are subtle, yet significant. The authors present very interesting results but missing much on providing mechanistic explainations.

I have a few comments and question:

There are a number of variables that may affect the results of the avoidance test and should be considered and mentioned:

1. Time for acclimation to the chamber - 5 minutes may be too short. Was this ever tested before?

2. The light intensity in natural environment of the medaka fish. If it is very different from the intensities used in the experiment, the results may be meaningless.

3. The light intensity in which the fish were raised prior to the experiments. If intensity changes dramatically it may have an effect.

Also, what do we see here? Is it the effect of the mutation on preception of light intesity? or the effect of the mutation on enxiety/stress?

Similarly, it is not clear what is the meaning of the change in behavior under blue light and dark and the age-dependent changes. Could the mutation have a developmental effect (see also below)? And, the subtle differences in the 48 hr experiment, could these reflect of change in chronotype? A change in chronotype could account for the differences in activity in the early morning. The answer to that, fish need to be followed under constant conditions. 

The paper describes an effect of tmt mutations on sst1b mRNA levels which was discovered by RNA-seq of mutant vs. wildtype brain regions. Information about this data is missing. For example, how many differentially expressed genes were found? What kind of genes? Was sst1b one of the highest changing genes?

The effect on sst1b is further analyzed by ISH on brain sections. This is a very important analysis. Looking at figure 5, it seems to me that there are less sst1b cells. Could this be a developmental effect? Are there other differentially expressed transcripts that are expressed in the same cell types? The answer may be hidden in the RNA-seq data and other possibly available scRNA-seq data. If the mutation has a developmental effect the results can and should be re-interpreted. 

The time of sampling in the photoperiod experiment is not clear. In page 11 it is stated that 'After one week, all fish were sacrificed between ZT0 and ZT3, respectively (blue arrowhead Fig. 6A)' Figure 6A shows that extending the day in long-day photoperiod doesn't change the time of ZT0. Therefore both groups could be sampled at the same time and the same ZT. Why '…ZT0 and ZT3, respectively'?

Regarding the presentation of the photoperiod experiment results. Are the results presented in 6C and 6D comparable? Are they normalized all together? It is important to know what has changed, was it lack of elevation upon placing in long photoperiod? 

Again, the same comparison of inoculated fish vs. untreated fish is important and it is not clear if they were all normilezed together. I would suggest that all results presented in figures 6C-E are normalized together. 

Discussion focuses of the photoperiod results and neglects much of the behavioral results. There should be much to discuss if the results are important.

Minor comments:

Introduction, 4th paragraph '….particularly to the gonads' I would suggest to change to 'particularly to the reproductive system'

Introduction, 5th paragraph, '….and represents ancestral ciliary-type Opsins'. Please explain what are C-opsins. It is a confusing term given that the c-opsins you are investigating here are not expressed in clasical ciliary photorecepto cells…. 

Page 6 second line, '….Fig. 1E,G,H), are statistically….' should be changed to '… Fig. 1E,G,H), became statistically….'

Pages 8-9, 'while in larval stages adding the tmt-opsin2 mutant ….. in the same experimental test during juvenile stages (Fig. 2G,H and Fig. 3I,J).' I do not see what is described - the slight effects are not much different from larva to juvenile stages. Am I missing something? And, again, the effect on activity may be an effect on chronotype.

In qPCRs, please indicate the number of samples.

Reviewer #2: TMT-Opsins differentially modulate medaka brain function in a context-dependent manner 

There is considerable interest in the role of the large opsin family in the regulation of cellular and animal physiology. However, part of the problem in this area is the large number of potential photopigments many of which have yet to be fully characterised. The authors present some experiments using the TALENs approach to attempt to knock down two members of the TMT family, tmt-opsin1b and tmt-opsin2. I was somewhat surprised that the authors fail to introduce the full complexity of the opsin genome in the teleosts. They investigate two of the three TMT families, with no clear rationale for their specific choice. The authors themselves have previously reported that one of the TMTs is co-expressed with at least one other non-visual opsin. This in itself alludes to some of the issues of ascribing a photopigment function to a single opsin. Thus, if you ascribe a photopigment function to a TMT – it could equally have an accessory function for another pigment, like for example VA-opsin? To assign a specific role to a particular pigment requires additional data. for example the spectral and absolute sensitivity of the system, and here this data is lacking.

The authors attempt to ascribe a specific function to two TMT opsins, but fail to address these opsins in sufficient detail to provide a full or definitive answer. There are also issues relating to the methodology and interpretation. In this regard the current data whilst interesting are still rather preliminary. In short they fail to provide a definitive and clear description of the role of TMT in the teleosts that would be suitable for a broader Biology audience. The study raises important issues that need to be addressed in significantly additional experiments.

Specific issues 

1) The authors do not fully consider potential additional off target effects in their TALENs approach and this may be important.

2) The authors fail to use stimuli that could identify the photopigment involved. TMT opsins have characteristic spectral sensitivity profile. However, the authors do not use monochromatic stimuli or a meaningful range of stimulus intensities in their experiments. Thus, the reader is left to question their functional origin. If the experiments were performed with conventional photobiology protocols the conclusions drawn would have a greater certainty. So for example, no evidence is provided to show that short wavelength light is more effective in the assays, which you might expect if TMT itself were responsible.

3) The authors do not establish that TMT is a short wavelength photopigment in any of the assays they describe – rather their data can only suggest they could have a role. This is a common and fundamental problem with a gene ablation approach in isolation. Ablating a TMT opsin does not in itself prove that it is a photopigment.

4) They present no convincing evidence for the proposed push-pull differential mechanism that they describe.

5) Importantly, given the previously published work, what would happen if the authors were to knock down VA opsin, which they themselves have shown to be co-expressed? 

Other issues

The sentence in the abstract as follows makes no sense: 

“We show that these Opsins interact non-additively, impacting the levels of the

preprohormone sst1b, as well as the voltage-gated sodium channel subunit scn12aa andat

least in part independently of the eyes and pineal- the amount of larval day-time rest.”

Reviewer #3: Fontinha et al conducted an original study where they report the effect of TMT-opsins (1b and 2) on Medaka brain's expression and behaviour. Using KO generated with TALEs, the authors show that these opsins modulate expression level of a somatostatin (sst1.1b) and the voltage gated sodium channel Nav1.9 (scn12aa) and influence daytime rest in Medaka larvae. 

This is a very ambitious endeavor.

Although the authors have a point that this topic of the non-visual functions of brain opsins is very interesting and surely not enough investigated, the manuscript overall does not bring a comprehensive view of the function of these 2 opsins on expression in the brain and relevance to behaviour. More information on anatomy, what is know on circuits expressing the opsins and the propeptide as well as the voltage gated sodium channel, is necessary to reach some understanding on the function of the opsins investigated. In addition, the manuscript is written in a convoluted manner, lacking both precision and clarity, which should be corrected in the revised version.

Major comments

1. Link between expression and function

As the authors found changes of expression of a pro-peptide and a sodium channel (downregulation of sst1.1b and scn12aa), does the corresponding modulation of these two genes explain the effect seen on the daytime rest ? and if yes, how? 

In order to reach a model on the function of these opsins in the Medaka larva, the authors should provide more data to tackle the problem:

a- the authors need to identify anatomically the brain regions where sst1.1b, tmt opsin 1b and scn12aa, tmt opsin 2 are expressed

b- based on the anatomical evidence and what is known in other species such as lamprey: do sst1.1 + cells receive projections from tmt opsin 1b + cells?

c- how do KO mutants for sst1.1 and scn12aa behave similarly to tmt opsin 1b and 2 KO? 

2. Behavioural characterization in the mutants:

a- Is there an effect on spontaneous locomotor activity independent of light levels?

When the authors present the avoidance assay at 7-8dpf, they argue that TMT-opsin 1b is expressed in tectum and reticular formation. A change in the activity of the reticular formation should be seen in basic exploration of Medaka larvae. Did the authors check at the same stage than the avoidance assay, that spontaneous locomotor activity was affected in mutant larvae ?

The authors report that at 9-12 dpf the locomotor activity of tmt 11b mutant was reduced while it was higher than siblings in the 20-22 dpf stage. This discrepancies could be explained by clutch to clutch variability more than real effects of the opsins. How many clutches were used to obtain each behavioral results ? These differences make us wonder if the effect of the mutation is light-dependent or just impacts independently of light the basal locomotor activity of the animal.

b- How can the authors explain that the tmt-opsin 1b mutants respond more in the avoidance assay than the control siblings ? are the cells expressing tmt opsin 1b inhibitory ? How does it work ?

3. Circuit underlying the behavioural effects:

- Nav1.9 has been involved with pain and excitability of cells while sst1.1 has been identified in numerous brain regions including habenula. Did the author check which brain nuclei are expressing sst1.1b and scn12aa at the same larval stage where the behavior experiments were conducted? Could it be that the same cells express sst1.1b and scn12aa and the tmt-opsins ? Could they be receiving inputs from them? We d like to know more about the circuits being involved with the integration of light and the output behavior, here: daytime rest.

- tmt opsin 2 complements the effect of tmt opsin 1b only at the larval but not juvenile stage: how does this work? where are these opsins express in larval and juvenile stages? 

Overall we need more precision in the information (anatomical identification of cells expressing, knowledge on the function of the nuclei targeted, .. ) to reach a comprehensive view. As it is the study provides only scattered information, which are not helping us understand how brain opsins modulate behaviour.

Minor comments

- Intro 2nd para - ref 2 to 9 : detail which behaviour and how the opsin contribute with more details and information, this paragraph is key for our understanding and it lacks content in its current form.

- Intro 5th para ref 19-24: provide more information on the opsin investigated and the actual effect on behavior with more precision again. 

- In the last paragraph of the introduction, the authors refer to the voltage-gated scn12aa as a neurotransmitter receptor for Nav1.9, which is completely wrong : ?

- Results: convoluted writing, effects should be clearly explained specifying direction and amplitude in every section.

---

## [Decision Letter · Decision Letter 2]

29 Oct 2020

Dear Kristin,

Thank you for submitting your revised Research Article entitled "TMT-Opsins differentially modulate medaka brain function in a context-dependent manner" for publication in PLOS Biology. I have now obtained advice from the original reviewer 1 and have discussed your revision as well with the Academic Editor. As mentioned previously, reviewers 2 and 3 could not re-review. I have included the comments from the Academic Editor below.

I'm delighted to let you know that we're now editorially satisfied with your manuscript. However before we can formally accept your paper and consider it "in press", we also need to ensure that your article conforms to our guidelines. A member of our team will be in touch shortly with a set of requests. As we can't proceed until these requirements are met, your swift response will help prevent delays to publication. Please also make sure to address the data and other policy-related requests noted at the end of this email.

- a cover letter that should detail your responses to any editorial requests, if applicable

*Copyediting*

*Published Peer Review History*

*Early Version*

Sincerely,

Gabriel Gasque, Ph.D.,

Senior Editor,

ggasque@plos.org,

PLOS Biology

DATA POLICY:

-- Please update your data files to include data for Figures 1H, 2BDHJLO, 3BDH, 4HJ, 5BCDE, 7CDE, S4CEGI, and S5A-T.

-- Please ensure that all data files are uploaded as 'Supporting Information' and are invariably referred to (in the manuscript, figure legends, and the Description field when uploading your files) using the following format verbatim: S1 Data, S2 Data, etc. Multiple panels of a single or even several figures can be included as multiple sheets in one excel file that is saved using exactly the following convention: S1_Data.xlsx (using an underscore).

-- Please also ensure that each figure legend in your manuscript includes information on where the underlying data can be found and that your supplemental data file/s has/have a legend.

Reviewer remarks:

Reviewer #1: The reviewers addressed all my comments.

Academic Editor: I am in total disagreement with reviewer 2, the comments that are mentioned are extremely hard and some of them are not reasonable to expect from a first study. I feel that the authors did a very good job addressing the comments considering the Covid situation and the excitement of the study. Finding that two ciliary opsins have critical role in light-mediated behaviors despite the large number of opsins in zebrafish is very important and shows that redundancy is not the reason that many opsins are expressed but that individual attributes to each opsin function and expression are important for the behavior. This gels quite nicely with the diversity we find in ipRGCs in the mammalian retina and adds to the importance of understanding that light effects that are not vision dependent are also elaborate and are not a simple on or off response. I therefore, believe strongly that we should publish this paper.

Let me provide you some of the comments from all reviewers that are quite harsh:

1- The authors should use natural light conditions: Not a single lab in the world has the ability to provide natural light conditions for their animals. This is really hard to require.

2- The authors should study VA opsin: What? They have already knocked out two opsins and have a functional result that is small but robust and significant. Asking them to add another is way outside the scope of the study.

3- Mechanistic insight and why: This would be a huge huge huge understanding for any study. And requesting this from the authors who found complementarity and not redundancy is unfair.

---

## [Editor Report · Decision Letter 3]

10 Dec 2020

Dear Dr. Tessmar-Raible,

I am writing concerning your manuscript submitted to PLOS Biology, entitled “TMT-Opsins differentially modulate medaka brain function in a context-dependent manner.”

We have now completed our final technical checks and have approved your submission for publication. You will shortly receive a letter of formal acceptance from the editor.

Kind regards,

PLOS Biology